# GROTHENDIECK GRAPH NEURAL NETWORKS FRAMEWORK: AN ALGEBRAIC PLATFORM FOR CRAFTING TOPOLOGY-AWARE GNNS

## ABSTRACT

Due to the structural limitations of Graph Neural Networks (GNNs), particularly those relying on conventional neighborhoods, alternative aggregation strategies have been explored to enhance GNN expressive power. This paper proposes a novel approach by generalizing the concept of neighborhoods through algebraic covers to overcome these limitations. We introduce the Grothendieck Graph Neural Networks (GGNN) framework, providing an algebraic platform for systematically defining and refining diverse covers for graphs. The GGNN framework translates these covers into matrix representations, extending the scope of designing GNN models by incorporating desired message-passing strategies. Based on the GGNN framework, we propose Sieve Neural Networks (SNN), a new GNN model that leverages the notion of sieves from category theory. SNN demonstrates outstanding performance in experiments, particularly in differentiating between strongly regular graphs, and exemplifies the versatility of GGNN in generating novel architectures.

## 1 INTRODUCTION

Where is the birthplace of the concept of neighborhood for nodes? Does this birthplace have the potential to generate other concepts as alternatives to neighborhoods to improve the expressive power of Graph Neural Networks (GNNs)? Due to their inherited reasons, most of existing GNN methods currently rely on neighborhoods as the foundation for message passing Gilmer et al. (2017). Several reasons support this preference. First, neighborhoods provide comprehensive coverage of graphs, encompassing all edges and directions, ensuring the entire graph participates in the message-passing process. Second, working with neighborhoods is straightforward, facilitated by the adjacency matrix. However, the localized perspective obtained from neighborhoods may result in shortcomings in GNN methods, such as their limited expressive power, which is at most equivalent to that of the Weisfeiler-Lehman (WL) test Sato (2020), Xu et al. (2019).

Extending the concept of neighborhoods or finding alternatives has been proposed as a way to address these limitations. In this regard, the topological characteristic of graphs has motivated the use of algebraic topology concepts. These concepts enable the examination of graphs from various perspectives, such as dimensions, faces, and boundaries, to capture higher-order interactions Bodnar et al. (2021b), Bodnar et al. (2021a). Furthermore, analyzing specific patterns and subgraphs provides the means for recognizing substructures as alternatives to neighborhoods Bouritsas et al. (2023), Ai et al. (2022). However, since neighborhoods are derived from a precise definition rather than a specific pattern, they cannot be represented effectively by the patterns.

We advocate that the algebraic viewpoint aligns more closely with the inherent nature of neighborhoods than the patterns. In other words, neighborhoods, emerging from the connections between edges, can be conceptualized as outcomes of an algebraic operation on edges. In this paper, we aim to algebraically extend the concept of neighborhoods in a way that not only enhances efficiency compared to neighborhood but also maintains simplicity of use. To this end, we explore the close relationship between category theory MacLane (1978) and graphs. We also observe that methods used to construct covers in category theory can serve as a schema for similar developments in graph

theory, where the concept of cover becomes meaningful with Grothendieck topologies MacLane & Moerdijk (1994).

Our contributions in this paper can be summarized as follows. First, we introduce the Grothendieck Graph Neural Networks (GGNN) framework, based on our interpretation of the Grothendieck topology, to establish the context for defining the concept of covers for graphs, and then transforming them into matrix forms for the message-passing process. The concept of covers in the GGNN framework differs from the traditional view of graphs based on neighborhoods, enabling alternative perspectives of the graph. In our proposed GGNN framework, a monoid $\mathrm{Mod}(G)$, generated by directed subgraphs, is introduced as the birthplace for the concept of neighborhoods, providing us the ability to generate various algebraic covers as alternatives to traditional neighborhood covers. Based on the proposed GGNN framework, we design a novel GNN model called Sieve Neural Networks (SNN), in which a graph $G$ is covered by a collection of elements from $\mathrm{Mod}(G)$, analogous to sieves in category theory MacLane & Moerdijk (1994).

## 2 GROTHENDIECK GRAPH NEURAL NETWORKS FRAMEWORK

In this section, we will move step by step to give meaning to the concept of cover for graphs and interpret them as matrices. With this, the necessary materials will be in hand to introduce the GGNN framework. By defining the matrix representation for a directed subgraph, we will provide a one-to-one correspondence between them, turning it into a monoidal homomorphism by introducing monoids generated by directed subgraphs and matrix representations. It will be proved that this monoidal homomorphism is invariant up to isomorphism and gives an algebraic description of a graph that will be the basis of our framework.

### 2.1 MATRIX REPRESENTATIONS OF DIRECTED SUBGRAPHS

This paper deals with undirected graphs; every graph has a fixed order on its set of nodes. We start by defining directed subgraphs and their matrix representations. Let $G = (V, E)$ be an undirected graph with $V$ as the set of nodes, $E$ as the set of edges, and a fixed order on $V$.

**Definition 2.1.1.** *(1) A path $p$ from node $v_{p_1}$ to node $v_{p_m}$ is an ordered sequence $v_{p_1}, e_{p_1}, v_{p_2}, e_{p_2}, \cdots, v_{p_{m-1}}, e_{p_{m-1}}, v_{p_m}$, where $e_{p_i}$ represents an edge connecting nodes $v_{p_i}$ and $v_{p_{i+1}}$.*

*(2) A directed subgraph $D$ of $G$ is a connected and acyclic subgraph of $G$ in which every edge of $D$ has a direction.*

A neighborhood is essentially a directed subgraph formed by combining directed edges leading to a specific node, see Figure 4. Using the adjacency matrix, we can represent each neighborhood with a matrix. In this representation, each column of the adjacency matrix corresponds to the neighborhood of the respective node. To isolate the representation of that specific neighborhood, we set the rest of the columns to zero. In the following definition, we expand this matrix representation to encompass directed subgraphs as a more general concept.

**Definition 2.1.2.** *For a directed subgraph $D$ of $G = (V, E)$, we define:*

*(1) $\leq_D$ to be a relation on $V$ in which $v_i \leq_D v_j$ if, based on the directions of $D$, there is a path in $D$ starting with $v_i$ and ending at $v_j$.*

*(2) the matrix representation for a directed subgraph $D$ to be a $|V| \times |V|$ matrix in which the entry $ij$ is 1 if $v_i \leq_D v_j$ and 0 otherwise.*

**Proposition 2.1.1.** *The relation $\leq_D$ is transitive.*

As stated in Definition 2.1.2, it is emphasized that a path within a directed subgraph must adhere to the directions. Directed subgraphs, viewed as a broader concept than neighborhoods, can be regarded as strategies for effectively broadcasting messages within a graph (see Figure 1). These subgraphs establish specific paths for message propagation, offering alternatives to connections based on neighborhoods. The matrix representation of a directed subgraph serves as a practical realization of the directed subgraph, enabling the implementation of the strategy derived from it. Consequently,

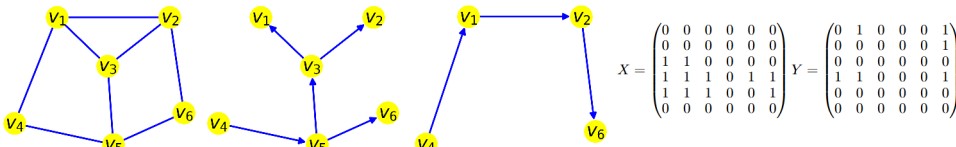

Figure 1: Here are two examples of directed subgraphs, $\hat{D}$ in the middle and $\bar{D}$ on the right, of a graph $G$ on the left. $X$ and $Y$ are the matrix representations of $\hat{D}$ and $\bar{D}$ respectively. The directed subgraphs $\hat{D}$ and $\bar{D}$ can be considered as strategies to broadcast the messages in the graph $G$, and their matrix representations make these strategies implementable.

matrix representations can be considered as substitutes for traditional adjacency matrices. The definition of matrix representation gives a map from the set of all directed subgraphs of $G$, denoted by $\mathsf{DirSub}(G)$, to the set of $|V| \times |V|$ matrices, denoted by $\mathsf{Mat}_{|V|}(\mathbb{R})$. Taking $\mathsf{MatRep}(G)$ as the image of this map, we get the following surjective function.

$$\mathsf{Rep} : \mathsf{DirSub}(G) \to \mathsf{MatRep}(G)$$

The following theorem shows the uniqueness of matrix representations for directed subgraphs. So, every directed subgraph can be determined completely by its representation.

**Theorem 2.1.1.** $\mathsf{Rep}$ *is an isomorphism.*

## 2.2 COVERING A GRAPH

While it is possible to cover a graph $G$ with a collection of elements from $\mathsf{DirSub}(G)$, and their matrix interpretation is accessible through $\mathsf{Rep}$, it is important to note that $\mathsf{DirSub}(G)$ is relatively small and lacks interaction among its elements. For example, the combination of $\hat{D}$ and $\bar{D}$, directed subgraphs presented in Figure 1, does not constitute a directed subgraph due to the presence of multiple paths between nodes. Consequently, its matrix interpretation does not exist, hindering its implementation in a message-passing process. This limitation poses challenges in designing diverse and meaningful strategies for message passing. To overcome this limitation and generate a more comprehensive set of elements, a method for combining them is required. Aiming for a broader space involves identifying an algebraic operation on $\mathsf{DirSub}(G)$. Pursuing a monoidal structure for $\mathsf{DirSub}(G)$, a common approach when transforming a set into an algebraic structure, seems appropriate. The operation $\bigoplus$ defined as follows can be a candidate. For $C, D \in \mathsf{DirSub}(G)$, the directed multigraph $C \bigoplus D$ is formed by taking the union of the sets of nodes and the disjoint union of the sets of directed edges. Thus, we have the commutative monoid $(\mathsf{Mult}(G), \bigoplus)$, where $\mathsf{Mult}(G)$ is defined as follows:

$$\mathsf{Mult}(G) = \{\bigoplus_{i=1}^{k} D_i : \text{for some } k \text{ and } D_1, \dots, D_k \in \mathsf{DirSub}(G)\}$$

In a multigraph, where multiple edges can exist between two nodes, the edges traversed by a path become crucial for specifying that path. This is why we highlight the edges between nodes in Definition 2.1.1. Consequently, this definition of a path is applicable to multigraphs as well. The combination of two directed subgraphs using the $\bigoplus$ operator results in an element that lacks substantial inheritance from its generators. The paths within the two generators play a limited role in determining the paths of the resulting element. Instead, the generated element has all paths formed by concatenating directed edges from its generators. Consequently, the $\bigoplus$ operation exhibits limited capability to generate innovative strategies for message passing. To address this, there is a need for an operation that demonstrates heightened sensitivity towards the paths within directed subgraphs. The subsequent theorem introduces a monoidal operation that extends beyond $\bigoplus$ and emphasizes the pivotal role of paths in strategy development. In this theorem and also throughout this paper, we are influenced by category theory, choosing to use the term *composition* instead of *concatenation* when referring to the amalgamation of two paths.

**Theorem 2.2.1.** *Let* $\mathsf{SMult}(G) = \{(M, S) : M \in \mathsf{Mult}(G), S \subseteq \mathsf{Paths}(M)\}$ *and the operation* $\bullet$ *be defined on* $\mathsf{SMult}(G)$ *as* $(M, S) \bullet (N, T) = (M \bigoplus N, S \star T)$, *where* $S \star T$ *is the union of the*

*sets $S, T$, and the collection of paths constructed by the composition of paths in $S$ followed by paths in $T$. Then $(\mathsf{SMult}(G), \bullet)$ is a non-commutative monoid.*

Note that in the above theorem, since the directed edges of $M \bigoplus N$ are obtained from the disjoint union of the directed edges of $M$ and $N$, the sets $S$ and $T$ are disjoint sets of paths as the subsets of $\mathsf{Paths}(M \bigoplus N)$. The operation $\bullet$ that acts as composition in categories allows the creation of the elements with allowed paths as desired. Non-commutativity of this operation comes from the composition of paths in $\star$. If $(M, S), (N, T) \in \mathsf{SMult}(G)$ do not have composable paths, then $\star$ is reduced to the union of $S$ and $T$, hence $(M, S) \bullet (N, T) = (N, T) \bullet (M, S)$. Considering $(M, S)$ in $\mathsf{SMult}(G)$ as a strategy for message passing, a multigraph is determined by $M$, and $S$ provides information about the allowed paths in $M$ for transferring messages. This monoid appears to be the appropriate place to define a cover as a collection of message-passing strategies. However, not all elements can be transformed into matrix form through an extension of Rep, and as a result, implementing strategies becomes challenging. To address this, we focus on selecting elements that can be transformed. By leveraging the fact that the set $\mathsf{DirSub}(G)$ can be embedded in $\mathsf{SMult}(G)$ by associating a directed subgraph $D$ with $(D, \mathsf{Paths}(D))$, we construct a suitable monoid for our objectives:

**Definition 2.2.1.** *For a graph $G = (V, E)$, we define the monoid of the directed subgraphs of $G$ to be the submonoid of $\mathsf{SMult}(G)$ generated by $\mathsf{DirSub}(G)$ and denote it by $\mathsf{Mod}(G)$.*

Hence, for an object $(M, S)$ in $\mathsf{Mod}(G)$, there are some directed subgraphs $D_1, \ldots, D_k$ of $G$ such that $(M, S) = D_1 \bullet \cdots \bullet D_k$ and so $M = \bigoplus_{i=1}^{k} D_i$ and $S = \mathsf{Paths}(D_1) \star \cdots \star \mathsf{Paths}(D_k)$. In the upcoming subsection, we aim to demonstrate that all elements belonging to the monoid $\mathsf{Mod}(G)$ can be transformed into matrix forms through an extension of Rep. This characteristic makes the monoid a valuable tool in achieving our goal of assigning meaning to the concept of covers for graphs. We define a cover for a graph as follows:

**Definition 2.2.2.** *A cover for a graph $G$ is a collection of finitely many elements of $\mathsf{Mod}(G)$.*

A cover, as defined, specifies a view of the graph and establishes some rules for its internal interactions. Within each element of $\mathsf{Mod}(G)$ lies a set of allowed paths that describe a localized strategy in transfers, and a cover as a collection of them can be seen as a collection of traffic rules. Benefiting from $\bullet$, the elements of a cover are capable of being integrated as well as interacting with each other. We have not mentioned in the definition that a cover must coat all nodes or edges. This gives the flexibility to select a cover suitable for a desired task.

With infinitely many elements and the noncommutative monoidal operation, $\mathsf{Mod}(G)$ greatly increases our ability to convert different perspectives and message-passing strategies to the covers. Also, the following theorem confirms the simplicity of making arbitrary elements and shows that the set of all directed edges generates $\mathsf{Mod}(G)$, so these elements together with the monoidal operation $\bullet$ are enough to construct suitable elements of the monoid $\mathsf{Mod}(G)$ to use in a cover. The cover presented in the next section exemplifies applying this theorem. Before stating the theorem, we show how the non-commutativity of $\bullet$ yields different elements by presenting a simple example.

**Example 2.2.1.** *For directed edges $d : u \to v$ and $e : v \to w$, the elements $d \bullet e$ and $e \bullet d$ are distinct. While both share the same directed edges in $d \bigoplus e$ as illustrated in $u \xrightarrow{\ d\ } v \xrightarrow{\ e\ } w$, they differ in terms of allowed paths. $d \bullet e$ includes a path from $u$ to $w$, whereas $e \bullet d$ lacks such a path. This highlights that the order of composition matters, resulting in different sets of allowed paths.*

**Theorem 2.2.2.** *Directed edges generate $\mathsf{Mod}(G)$.*

## 2.3 MATRIX INTERPRETATION OF A COVER

In this section, our objective is to extend the morphism Rep to a monoidal homomorphism, encompassing $\mathsf{Mod}(G)$ as its domain. This extension plays a pivotal role in the GGNN framework, transforming a cover into a collection of matrices. Since $\mathsf{Mod}(G)$ extends $\mathsf{DirSub}(G)$, we aim to move beyond $\mathsf{MatRep}(G)$ and enter a broader realm where matrix transformations corresponding to elements of $\mathsf{Mod}(G)$ reside. At first, we define the binary operation $\circ$ on $\mathsf{Mat}_n(\mathbb{R})$, the set of all $n \times n$ matrices, as follows:

$$A \circ B = A + B + AB$$

**Theorem 2.3.1.** $(\mathsf{Mat}_n(\mathbb{R}), \circ)$ *is a monoid.*

Now we are ready to extend $\mathsf{MatRep}(G)$ to a monoid:

**Definition 2.3.1.** *The monoid of matrix representations of a given graph $G = (V, E)$ is defined to be the submonoid of $(\mathsf{Mat}_{|V|}(\mathbb{R}), \circ)$ generated by $\mathsf{MatRep}(G)$, denoted by $(\mathsf{Mom}(G), \circ)$.*

To define a monoidal homomorphism between the monoids $(\mathsf{Mod}(G), \bullet)$ and $(\mathsf{Mom}(G), \circ)$ in such a way that it is an extension of the morphism $\mathsf{Rep}$, we need the following theorem which gives a good explanation of the monoidal operation $\circ$.

**Theorem 2.3.2.** *For $A_1, A_2, \cdots, A_k \in \mathsf{Mat}_n(\mathbb{R})$ with $k \in \mathbb{N}$ we have:*

$$A_1 \circ A_2 \circ \cdots \circ A_k = \sum_{i=1}^{k} A_i + \sum_{\sigma \in O(k,2)} A_{\sigma_1} A_{\sigma_2} + \cdots + \sum_{\sigma \in O(k,j)} A_{\sigma_1} \cdots A_{\sigma_j} + \cdots + A_1 A_2 \cdots A_k$$

*where $O(k, i)$ is the set of all strictly monotonically increasing sequences of $i$ numbers of $\{1, \cdots, k\}$*

Now, we present the extension of $\mathsf{Rep}$ as a monoidal homomorphism, mapping elements of $\mathsf{Mod}(G)$ to elements of $\mathsf{Mom}(G)$ while preserving the monoidal operations.

**Theorem 2.3.3.** *The mapping $\mathsf{Tr} : \mathsf{Mod}(G) \longrightarrow \mathsf{Mom}(G)$*

$$(M, S) = D_1 \bullet D_2 \bullet \cdots \bullet D_k \longmapsto A = A_1 \circ A_2 \circ \cdots \circ A_k$$

*is a surjective monoidal homomorphism, where $D_i \in \mathsf{DirSub}(G)$ and $A_i = \mathsf{Rep}(D_i)$.*

We refer to $\mathsf{Tr}(M, S)$ as the matrix transformation of $(M, S)$. In the proof of Theorem 2.3.3, it becomes evident that $\mathsf{Tr}$ functions as a path counter, assigning the number of paths in $S$ between two nodes $v_i$ and $v_j$ to the entry $ij$ of the matrix $\mathsf{Tr}(M, S)$. This monoidal surjection interprets covers as collections of matrices, establishing a relationship similar to that between the adjacency matrix and neighborhoods. While our attempts to establish $\mathsf{Tr}$ as an isomorphism have not succeeded, its nature as an extension of an isomorphism, coupled with its ability to characterize a graph up to isomorphism (as we will show in the next subsection), reinforces the validity of the matrix transformations derived from it for covers. Given the surjective nature of $\mathsf{Tr}$, we have:

**Corollary 2.3.1.** *Matrix representations of directed edges generate $\mathsf{Mom}(G)$.*

**Example 2.3.1.** *In Figure 1, two directed subgraphs, $\hat{D}$ and $\bar{D}$, of a graph $G$ are illustrated with their respective matrix representations, denoted as $X$ and $Y$. We highlighted that these subgraphs can be viewed as strategies for broadcasting messages within the graph. Through the operation $\bullet$, we can combine them to form new strategies, $\hat{D} \bullet \bar{D}$ and $\bar{D} \bullet \hat{D}$. Utilizing the matrix transformations obtained via $\mathsf{Tr}$, we can implement these combined strategies for the message-passing process.*

$$\mathsf{Tr}(\hat{D} \bullet \bar{D}) = \mathsf{Tr}(\hat{D}) \circ \mathsf{Tr}(\bar{D}) = X \circ Y \text{ and } \mathsf{Tr}(\bar{D} \bullet \hat{D}) = \mathsf{Tr}(\bar{D}) \circ \mathsf{Tr}(\hat{D}) = Y \circ X$$

## 2.4 ALGEBRAIC DESCRIPTION OF A GRAPH

So far, for an arbitrary graph, two monoids and a monoidal homomorphism between them have been presented. The question that arises now is *how much these monoidal structures can describe a graph*. To answer this question, some preliminaries are needed. We define a special type of linear isomorphism between vector space of matrices. A matrix $A \in \mathsf{Mat}_n(\mathbb{R})$ is actually a linear transformation from $\mathbb{R}^n$ to itself. Reordering the standard basis of $\mathbb{R}^n$ changes the matrix representation of the linear transformation in such a way that it will be obtained by reordering rows and columns of matrix $A$. These actions change the indices of entries of $A$; so a change in the order of the standard basis of $\mathbb{R}^n$ gives a linear isomorphism from $\mathsf{Mat}_n(\mathbb{R})$ to itself. We call this kind of linear isomorphism a **Change-of-Order mapping**, see Example B.0.1. The Change-of-Order mappings are compatible with the monoidal structure of $\mathsf{Mat}_n(\mathbb{R})$ as shown in the following proposition:

**Proposition 2.4.1.** *Suppose $f : \mathsf{Mat}_n(\mathbb{R}) \to \mathsf{Mat}_n(\mathbb{R})$ is a Change-of-Order mapping. Then $f$ preserves $\circ$, matrix multiplication and element-wise multiplication.*

Now, we want to investigate the effect of two isomorphic graphs on their corresponding monoidal structures and vice versa. A graph isomorphism $f : G \to H$ is a change in the chosen order of

the nodes. So it induces a Change-of-Order mapping $\mathsf{CO}(f) : \mathsf{Mat}_{|V_G|}(\mathbb{R}) \to \mathsf{Mat}_{|V_H|}(\mathbb{R})$. The following theorem shows that isomorphic graphs have isomorphic monoidal structures described in Theorem 2.3.3.

**Theorem 2.4.1.** *Every graph isomorphism $f : G \to H$ induces monoidal isomorphisms $\mathsf{Mod}(f) : \mathsf{Mod}(G) \longrightarrow \mathsf{Mod}(H)$ and $\mathsf{Mom}(f) : \mathsf{Mom}(G) \to \mathsf{Mom}(H)$ such that the Diagram 1 is commutative, where $\iota$ represents the inclusions.*

$$
\begin{array}{ccccc}
\mathsf{Mod}(G) & \xrightarrow{\mathsf{Tr}_G} & \mathsf{Mom}(G) & \xrightarrow{\iota} & \mathsf{Mat}_{|V_G|}(\mathbb{R}) \\
{\scriptstyle\mathsf{Mod}(f)}\downarrow & & {\scriptstyle\mathsf{Mom}(f)}\downarrow & & \downarrow{\scriptstyle\mathsf{CO}(f)} \\
\mathsf{Mod}(H) & \xrightarrow[\mathsf{Tr}_H]{} & \mathsf{Mom}(H) & \xrightarrow[\iota]{} & \mathsf{Mat}_{|V_H|}(\mathbb{R})
\end{array}
\tag{1}
$$

The converse of Theorem 2.4.1 can be stated as follows:

**Theorem 2.4.2.** *Suppose $G$ and $H$ are two graphs with $|V_G| = |V_H| = n$, and $f : \mathsf{Mat}_n(\mathbb{R}) \to \mathsf{Mat}_n(\mathbb{R})$ is a Change-of-Order mapping. If the restriction of $f$ to $\mathsf{Mom}(G)$ yields an isomorphism to $\mathsf{Mom}(H)$, then $G$ and $H$ are isomorphic.*

## 2.5 DEFINITION OF THE GGNN FRAMEWORK

Theorems 2.4.1 and 2.4.2 lay the foundation for our framework. These theorems establish that graphs $G$ and $H$ are isomorphic if and only if the vertical homomorphisms in Diagram 1 are isomorphisms. This crucially implies that altering the node order in a graph induces isomorphic changes in both a cover and its matrix interpretation. Thus, the horizontal homomorphisms in Diagram 1 serve as a complete determination of graphs, providing algebraic descriptions for them. Leveraging this diagram, we define the GGNN framework as follows:

**Definition 2.5.1.** *The Grothendieck Graph Neural Networks framework for a graph $G = (V, E)$ is defined to be the algebraic description:*

$$
\mathsf{Mod}(G) \xrightarrow{\mathsf{Tr}} \mathsf{Mom}(G) \xrightarrow{\iota} \mathsf{Mat}_{|V|}(\mathbb{R})
\tag{2}
$$

The GGNN framework introduces various actions for creating, translating, and enriching a cover. $\mathsf{Mod}(G)$ offers multiple choices of covers, serving as alternatives for cover of neighborhoods. The transformation $\mathsf{Tr}$ converts these covers into collections of matrices, and, with the mapping $\iota$, these collections are transported to a larger space, providing an opportunity to enrich them using elements of $\mathsf{Mat}_{|V|}(\mathbb{R})$ and the allowed operation presented in Proposition 2.4.1. For more details see D. As promised, the following theorem demonstrates that the GGNN framework can indeed be considered the birthplace of neighborhoods:

**Theorem 2.5.1.** *The collection of neighborhoods, which forms the basis for MPNNs, constitutes a cover in the context of the GGNN framework and can be transformed into an adjacency matrix.*

The GGNN framework provides the ability to create a cover through a precise definition that can be applied to any arbitrary graph, similar to the definition of neighborhoods. In the next section, we illustrate this capability by presenting a cover constructed using the precise definition of certain elements of $\mathsf{Mod}(G)$.

## 3 SIEVE NEURAL NETWORK, A MODEL BASED ON GGNN FRAMEWORK

Based on the concept of sieves in category theory, we will introduce a GNN model, called *Sieve Neural Networks* (SNN), which will be constructed in the GGNN framework. In this model, each node spreads its roots in the graph like a growing seed and tries to feed itself through these roots. We mean this story by creating appropriate elements of $\mathsf{Mod}(G)$ for a graph $G$ and considering their collection as a cover for the graph. The connections resulting from this cover provide various ways for message passing between nodes that lead to a knowledge of the graph topology. In the following, we explain the process of creating the desired cover and introduce the model based on them.

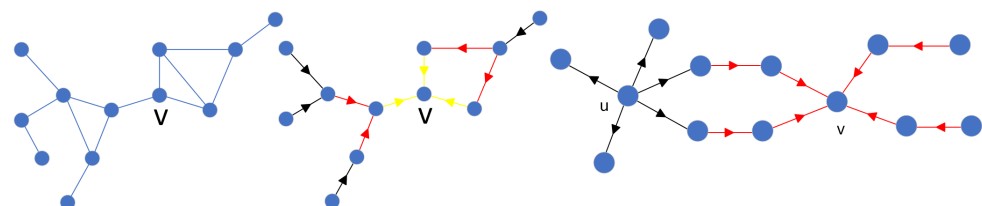

Figure 2: Left: A graph $G$. Middle: $\mathsf{Sieve}(v,3) = D_3(v) \bullet D_2(v) \bullet D_1(v)$ for $v \in G$. Directed edges in yellow, red and black specify $D_1(v)$, $D_2(v)$ and $D_3(v)$ recpectively. Right: A graph $H$. $\mathsf{CoSieve}(u,1) \bullet \mathsf{Sieve}(v,2)$ as an element of $\mathsf{Mod}(H)$ determines the ways of establishing contact between $u$ and $v$ in $\mathsf{SNN}(\alpha,(1,2))$

### 3.1 CONSTRUCTING THE MODEL

**Generating the desired elements of** $\mathsf{Mod}(G)$: For node $v$, we create the sets $M_k(v)$ as follows:

$*$ $N_0(v) = \{v\}$, $M_0 = \emptyset$

$*$ $N_1(v) = N(v)$, its neighborhood, $M_1(v) = \{w \to u : wu \in E, w \in N_1(v), u \in N_0(v)\}$

$*$ and inductively for $k \in \mathbb{N}$,

$$N_k(v) = \bigcup_{u \in N_{k-1}(v)} N(u) - \bigcup_{i=0}^{k-1} N_i(v), \; M_k(v) = \{w \to u : wu \in E, w \in N_k(v), u \in N_{k-1}(v)\}$$

The directed edges in $M_i(v)$ are not composable. Therefore, disregarding the order and non-commutativity of $\bullet$, we define $D_i(v) := \bullet e_{e \in M_i(v)}$ and utilizing them to generate the elements

$$\mathsf{Sieve}(v,k) := D_k(v) \bullet D_{k-1}(v) \bullet \cdots \bullet D_1(v) \bullet D_0(v)$$

of $\mathsf{Mod}(G)$ in which $D_0(v)$ is the identity of $\mathsf{Mod}(G)$, see Figure 2. Obviously there is some $k_0$ such that $M_{k_0} \neq \emptyset$ and $\emptyset = M_{k_0+1} = M_{k_0+2} = \cdots$. Then $\mathsf{Sieve}(v,k_0) = \mathsf{Sieve}(v,k_0+1) = \cdots$ We denote the element $\mathsf{Sieve}(v,k_0)$ by $\mathsf{Sieve}(v,-1)$. To construct the opposite of $\mathsf{Sieve}(v,k)$, we define $M_i^{op}(v)$ as the set containing the edges in $M_i(v)$ with the opposite directions. We then create $D_i^{op}(v) := \bullet e_{e \in M_i^{op}(v)}$. This results in new elements of $\mathsf{Mod}(G)$:

$$\mathsf{CoSieve}(v,l) := D_0^{op}(v) \bullet D_1^{op}(v) \bullet \cdots \bullet D_{l-1}^{op}(v) \bullet D_l^{op}(v)$$

.

**The Cover of Sieves**: So far, for a graph $G$, the following elements of $\mathsf{Mod}(G)$ have been selected for every node $v$. Our desired cover for a graph $G$ is the collection containing all these elements for all nodes in $G$ and we call it the cover of sieves.

$$\mathsf{Sieve}(v,0), \mathsf{Sieve}(v,1), \cdots, \mathsf{Sieve}(v,-1) \text{ and } \mathsf{CoSieve}(v,0), \mathsf{CoSieve}(v,1), \cdots, \mathsf{CoSieve}(v,-1)$$

**Matrix Interpretation of The Cover of Sieves**: The mapping $\mathsf{Tr}$ gives the matrix interpretation of the cover of sieves, transforming it into a collection of elements of $\mathsf{Mom}(G)$, denoted as follows:

$$\mathsf{Image}(v,k) := \mathsf{Tr}(\mathsf{Sieve}(v,k)), \quad \mathsf{CoImage}(v,l) := \mathsf{Tr}(\mathsf{CoSieve}(v,l))$$

Since $\mathsf{Tr}$ is a monoidal homomorphism, $\mathsf{Image}(v,k)$ can be expressed as follows, providing insight into its computational procedure:

$$\begin{aligned}\mathsf{Image}(v,k) = \mathsf{Tr}(\mathsf{Sieve}(v,k)) &= \mathsf{Tr}(D_k(v) \bullet D_{k-1}(v) \bullet \cdots \bullet D_0(v)) \\ &= \mathsf{Tr}(D_k(v)) \circ \mathsf{Tr}(D_{k-1}(v)) \circ \cdots \circ \mathsf{Tr}(D_0(v))\end{aligned} \quad (3)$$

Therefore, to calculate $\mathsf{Image}(v,k)$, it is necessary to transform $D_i(v)$ into matrix form. As we mentioned, directed edges in $M_i(v)$ are not composable. Then $\mathsf{Tr}(e)\mathsf{Tr}(c) = \mathsf{Tr}(c)\mathsf{Tr}(e) = 0$ for $e,c \in M_i(v)$. Theorem 2.3.2 implies:

$$\mathsf{Tr}(D_i(v)) = \mathsf{Tr}(\bullet e_{e \in M_i(v)}) = \circ \mathsf{Tr}(e)_{e \in M_i(v)} = \sum_{e \in M_i(v)} \mathsf{Tr}(e)$$

So obtaining $\mathsf{Tr}(D_i(v))$ is achievable from the adjacency matrix of $G$ based on the definition of $M_i$. It is easy to verify that $\mathsf{ColImage}(v,l)$ is the transpose of $\mathsf{Image}(v,l)$. Therefore, computing one of them is enough.

**Building The Model**: Based on the cover of sieves and its matrix interpretation, we present our model, SNN, with varying levels of complexity as follows:

**The version** $\mathsf{SNN}(\alpha,(l,k))$**:** In the $\alpha$ version of SNN, $\mathsf{Sieve}(v,k)$ is considered as a receiver and $\mathsf{CoSieve}(v,l)$ as a sender for a node $v$. For nodes $v_i$ and $v_j$, the ways to transmit information from $v_i$ to $v_j$ are the allowed paths from $v_i$ to $v_j$ in $\mathsf{CoSieve}(v_i,l) \bullet \mathsf{Sieve}(v_j,k)$ see Figure 2. The number of these paths is $ij$ entry of $\mathsf{ColImage}(v_i,l) \circ \mathsf{Image}(v_j,k)$. By dividing this number by the product of the summation of entries in the $i$-th row of $\mathsf{ColImage}(v_i,l)$ and the summation of entries in the $j$-th column of $\mathsf{Image}(v_j,k)$, we obtain the ratio of established paths between $v_i$ and $v_j$ to the maximum expected paths. The matrix resulting from performing this process for all $i$s and $j$s is the output of $\mathsf{SNN}(\alpha,(l,k))$ for graph $G$. The division step is a way for preserving additional information from $\mathsf{CoSieve}(v_i,l) \bullet \mathsf{Sieve}(v_j,k)$ in the model's output. Omitting this step results in denoting the model as $\mathsf{SNN}_o(\alpha,(l,k))$.

**The version** $\mathsf{SNN}(\beta,(l_1,\cdots,l_t))$**:** In the $\beta$ version of SNN, we leverage $\mathsf{Mat}_n(\mathbb{R})$ by mapping the collection of Images and Colmages into it. Here, additional operations become available, and we choose summation. For $l_i$, if $i$ is odd, we denote by $Su_i$ the summation (over nodes) of all matrices $\mathsf{ColImage}(v,l_i)$ and if $i$ is even, $Su_i$ is the summation of all matrices $\mathsf{Image}(v,l_i)$. Ultimately, the matrix $Su_1 \circ \cdots \circ Su_t$ represents the output of $\mathsf{SNN}(\beta,(l_1,\cdots,l_t))$ for graph $G$.

With versions $\alpha$ and $\beta$ of SNN, two approaches are introduced for integrating matrices from the matrix interpretation of the cover of sieves to form a unified matrix. Utilizing the relationship $\mathsf{ColImage}(v_i,l) = \mathsf{Image}(v_i,l)^{tr}$, we derive $\mathsf{ColImage}(v_i,l) \circ \mathsf{Image}(v_j,k) = (\mathsf{ColImage}(v_j,k) \circ \mathsf{Image}(v_i,l))^{tr}$. Consequently, the output of $\mathsf{SNN}(\alpha,(l,k))$ is the transpose of the output of $\mathsf{SNN}(\alpha,(k,l))$. This symmetry implies that the output of $\mathsf{SNN}(\alpha,(l,l))$ is symmetric as well. However, this symmetry doesn't hold for $\mathsf{SNN}(\alpha,(l,k))$ when $l \neq k$ (see Example B.0.2), and as a result, the outputs of $\mathsf{SNN}(\alpha,(l,k))$ and $\mathsf{SNN}(\alpha,(k,l))$ may differ in general. Nonetheless, a comparative analysis allows us to conclude that $\mathsf{SNN}(\alpha,(l',k'))$ can capture more paths than $\mathsf{SNN}(\alpha,(l,k))$ if $l \leq l'$ and $k \leq k'$.

Version $\beta$ of SNN is designed to offer a more comprehensive representation of the cover of sieves. The collection $\{\mathsf{Sieve}(v,l_i)\}$ (or $\{\mathsf{CoSieve}(v,l_i)\}$) forms a subcover within the cover of sieves. The matrix $Su_i$ can be seen as an interpretation of this subcover, representing all allowed paths for the elements within it. By the element $Su_1 \circ \cdots \circ Su_t$, we obtain a matrix that interprets a specific combination of these subcovers. This resulting matrix represents the paths formed by the composition of allowed paths from the mentioned subcovers, providing a distinct interpretation of the cover of sieves.

The output of SNN can be viewed as a weighted graph representation, suitable as input for various GNN methods that involve message passing. This implies that any GNN can leverage the cover of sieves instead of the traditional cover of neighborhoods. The following theorem establishes the invariance of SNN, highlighting its efficiency for utilization in graph classification tasks.

**Theorem 3.1.1.** SNN *is invariant.*

**SNN in Practice**: Most of the time, GNNs deal with featured graphs. In the case of SNN, Equation 3 is the point to incorporate edge features. For a featured graph $G = (V,E,F)$ with edge features in $\mathbb{R}^m$, replacing 1s with the corresponding edge features in $\mathsf{Tr}(D_i(v))$ yields a matrix where entries are sourced from $m$-dimensional vectors. Through the update operation $\circ$, employing element-wise summation and multiplication for $m$-dimensional vectors, SNN acquires the ability to deal with featured graphs. Additionally, by multiplying $\mathsf{Tr}(D_i(v))$s by a constant $\gamma \in (0,1]$, the model can be enhanced in a manner that is sensitive to the length of paths.

## 3.2 COMPARING WITH MPNN

For a node $v$, its neighborhood can be described by the element $\mathsf{Sieve}(v,1)$. Consequently, $\mathsf{SNN}_o(\alpha,(0,1))$ and $\mathsf{SNN}_o(\alpha,(1,0))$ correspond to the adjacency matrix, signifying their utilization of neighborhoods for message passing. This is equivalent to MPNNs. Hence, SNN can be

Table 1: Accuracy on TUD datasets. The top three are highlighted by First, Second, Third. *Graph Kernel Methods

| Dataset | MUTAG | PTC | NCI1 | IMDB-B | IMDB-M |
|---|---|---|---|---|---|
| WL kernel* Shervashidze et al. (2011) | 90.4±5.7 | 59.9±4.3 | 86.0±1.8 | 73.8±3.9 | 50.9±3.8 |
| GNTK* Du et al. (2019) | 90.0±8.5 | 67.9±6.9 | 84.2±1.5 | 76.9±3.6 | 52.8±4.6 |
| GIN Xu et al. (2019) | 89.4±5.6 | 64.6±7.0 | 82.7±1.7 | 75.1±5.1 | 52.3±2.8 |
| PPGNs Maron et al. (2019) | 90.6±8.7 | 66.2±6.6 | 83.2±1.1 | 73.0±5.8 | 50.5±3.6 |
| GSN Bouritsas et al. (2023) | 92.2±7.5 | 68.2±7.2 | 83.5±2 | 77.8±3.3 | 54.3±3.3 |
| TL-GNN Ai et al. (2022) | 95.7±3.4 | 74.4±4.8 | 83.0±2.1 | 79.7±1.9 | 55.1±3.2 |
| SIN Bodnar et al. (2021b) | N/A | N/A | 82.8 ± 2.2 | 75.6 ± 3.2 | 52.5 ± 3.0 |
| CIN Bodnar et al. (2021a) | 92.7 ± 6.1 | 68.2 ± 5.6 | 83.6 ± 1.4 | 75.6 ± 3.7 | 52.7 ± 3.1 |
| SNN | 96.11±3.3 | 77.3±4.1 | 83.6±1.2 | 80.5±3 | 54.53±2.23 |

considered as a generalization of MPNNs. In Example B.0.2, two graphs are considered that MPNN can not distinguish, yet SNN can. This example illustrates how a shift in perspective, resulting from a change in cover, reveals the topological properties of graphs.

### 3.3 COMPLEXITY

According to Equation 3, $\mathsf{Image}(v, k)$ can be computed by executing $k$ iterations matrix multiplication and summation. Consequently, the time complexity of it for all nodes is $\mathcal{O}(kn^4 + kn^3)$ where $n$ is the number of nodes. Assume $l \leq k$. Equation 3 implies:

$$\mathsf{Image}(v, k) = \mathsf{Tr}(D_k(v)) \circ \cdots \circ \mathsf{Tr}(D_{l+1}(v)) \circ \mathsf{Image}(v, l)$$

Therefore in the process of computing $\mathsf{Image}(v, k)$, we simultaneously obtain $\mathsf{Image}(v, l)$ for all $l \leq k$. As previously mentioned, $\mathsf{ColImage}(v, l)$ is the transpose of $\mathsf{Image}(v, l)$. Consequently, in computation of $\mathsf{SNN}(\alpha, (l, k))$, the time complexity of all $\mathsf{Image}(v, k)$ and $\mathsf{ColImage}(v, l)$ is $\mathcal{O}(kn^4 + kn^3 + n^3)$. The operations $\circ$ between Images and ColImages contributes $n^4 + n^3$ to the complexity, resulting in $\mathcal{O}((k+1)n^4 + kn^3 + 2n^3)$. Then the time complexity of $\mathsf{SNN}(\alpha, (l, k))$ is $\mathcal{O}(n^4)$. For version $\beta$ of the model, set $l_0 = Max(l_1, \cdots, l_t)$. The time complexity of $\mathsf{Image}(v, l_0)$ for all nodes is $\mathcal{O}(l_0 n^4 + l_0 n^3)$. Similar to version $\alpha$, computing ColImages adds $(t/2)n^3$ to the complexity. Additionally computing $Su_i$s and $Su_1 \circ Su_2 \circ \cdots \circ Su_t$ adds $2tn^3 + tn^2$ to the complexity, resulting in $\mathcal{O}(l_0 n^4 + (l_0 + (5/2)t)n^3 + tn^2)$ for time complexity. Therefore, the time complexity of $\mathsf{SNN}(\beta, (l_1, \cdots, l_t))$ is $\mathcal{O}(n^4)$. If the adjacency matrix is sparse, both cases can be reduced to $\mathcal{O}(|E| \cdot |V|^2)$ by leveraging sparse matrix operations.

### 3.4 EXPERIMENTS

In this section, we conduct a comprehensive evaluation of SNN across various datasets. In the first experiment, we assess SNN's capability to differentiate between graphs, providing a practical benchmark against the WL test. In this experiment, we employ robust versions $\beta$ of the model. In the second experiment, we extend the evaluation to classical datasets designed for graph classification. Here, considering the potential risk of overfitting, we adopt version $\alpha$ levels of the model that are slightly more potent than MPNN to ensure a smooth performance in the experiment. These experiments demonstrate the flexibility of SNN in handling diverse tasks and datasets.

**SR**: To assess the discriminative capability of SNN in identifying non-isomorphic graphs, we utilized all the publicly available collections of **Strongly Regular graphs** accessible at http://users.cecs.anu.edu.au/ bdm/data/graphs.html. Strongly Regular graphs pose challenges for graph isomorphism, given that the 3-WL test fails to conclusively differentiate pairs of such graphs Bodnar et al. (2021b). As SNN is invariant, our focus lies on the model's outputs for graphs. In this experiment, where overfitting is not discussed, we employed a potent level of SNN. By applying $\mathsf{SNN}(\beta, (-1, -1, -1))$ to graphs within each collection and computing $\mathsf{Mean}$ and $\mathsf{Var}$ on the output matrices and their diagonals, a 4-dimensional vector associated with each graph is obtained, forming an embedding. Given SNN's invariance, isomorphic graphs share identical embeddings. Our observations reveal that the model can effectively differentiate between all graphs within each collection.

**CSL**: We also evaluate SNN on the Circular Skip Link dataset (CSL) as a benchmark to assess the expressivity of GNNs Murphy et al. (2019), Dwivedi et al. (2023). CSL comprises 150 4-regular

graphs categorized into 10 different isomorphism classes. Applying $\mathsf{SNN}(\beta, (-1))$ to graphs in the dataset, we compute $\mathsf{Sum}$ on the resulting matrices, forming a function. Due to $\mathsf{SNN}$'s invariance, isomorphic graphs yield the same value. Our observations reveal that this variant of $\mathsf{SNN}$ successfully distinguishes the 10 different isomorphism classes, with graphs within the same class sharing identical values.

**TUD datasets**: We evaluate SNN on five datasets: **MUTAG**, **PTC**, **NCI1**, **IMDB-B**, and **IMDB-M** from the TUD benchmarks, comparing against various GNNs and Graph Kernels. For all datasets except **NCI1**, we employ $\mathsf{SNN}(\alpha, (1,1))$. Recognizing the need for a more complex version for **NCI1**, we utilize $\mathsf{SNN}(\alpha, (1,2))$. Both versions of SNN are slightly more potent than MPNN, equivalent to $\mathsf{SNN}(\alpha, (0,1))$. For datasets **MUTAG** and **PTC**, which have edge features, we replace 1s with the corresponding edge features in $\mathsf{Tr}(D_i(v))$, and in all cases, we enhance SNN by multiplying $\mathsf{Tr}(D_i(v))$ by the constant $\gamma = 0.5$ to increase sensitivity to the length of paths. We treat the output of SNN as a weighted graph for datasets lacking edge features and an edge-featured graph for datasets with edge features. We utilize GNN operators **GraphConv** and **GINEConv** provided by PyTorch Geometric Fey & Lenssen (2019), based on GNNs introduced in Morris et al. (2019) and Hu et al. (2020), respectively. Tenfold cross-validation is performed. In Table 1, we report the accuracies and compare them against a collection of Graph Kernels and GNNs. The results demonstrate that SNN has achieved good performance across this diverse set of datasets.

## 4 RELATED WORK

The research on improving Message Passing Neural Networks (MPNNs) in Graph Neural Networks (GNNs) focuses on enhancing the neighborhood-based message-passing process. Various methods aim to either transform the graph representation or augment node and substructure information to increase MPNN expressiveness. Gilmer et al. (2017) proposes that classical GNN methods can be unified under MPNNs. Many follow-up works aim to expand beyond simple neighborhood-based interactions. In Gasteiger et al. (2021), Directed Line Graphs (DirMPNN) replace the original graph with a directed line graph where nodes represent directed edges, enhancing message-passing accuracy. Ai et al. (2022) introduces Topology-aware GNNs (TLGNN), which use an additional visualization graph to capture structural features, enabling more informed message passing. In Bouritsas et al. (2023), Graph Substructure Networks (GSNs) analyze specific graph patterns to add structure-based features, while Feng et al. (2022a) introduces KerGNNs that use graph filters, inspired by convolutional neural networks, to capture local subgraphs for more precise node feature updates. Methods like You et al. (2021) and Feng et al. (2022b) focus on improving node representations by considering extended neighborhoods and ego networks, with the latter introducing new kernel-based methods for K-hop neighbor aggregation. Vignac et al. (2020) enhances node features by incorporating local context matrices that reflect a node's surrounding topology, improving tasks like cycle detection. The method in Papp et al. (2021) introduces random node removal with low probability, running MPNNs on slightly altered graphs to propagate results and preserve graph topology. These methods demonstrate varied strategies for making MPNNs more expressive and capable of capturing complex graph topologies.

## 5 CONCLUSION

In this paper, the concept of cover for graphs is defined as an algebraic extension of neighborhoods, and a novel framework is introduced that paves the way for the design of various models for GNN based on the desired cover. An algebraic platform for transforming the covers into collections of matrices adds to the simplicity of the framework's designed models. Also, based on this framework, we build a novel model for GNN, which makes working with the framework clearer, in addition to good results in experiments. Looking ahead, our future work aims to delve deeper into the power and potential applications of the GGNN framework. We plan to conduct a more comprehensive theoretical comparison between SNN and the Weisfeiler-Lehman test.

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

## A   DEFINITIONS

The definition of a monoid is as follows Hungerford (1980):

**Definition A.0.1.** *A monoid is a non-empty set* M *together with a binary operation* $\cdot$ *on* M *which*

1) *is associative:* $a \cdot (b \cdot c) = (a \cdot b) \cdot c$ *for all* $a, b, c \in \mathsf{M}$ *and*

2) *contains identity element* $e \in \mathsf{M}$ *such that* $a \cdot e = e \cdot a = a$

*If, for all* $a, b \in \mathsf{M}$*, the operation satisfies* $a \cdot b = b \cdot a$*, then we say that* $\mathsf{M}$ *is a commutative monoid.*

# B EXAMPLES

**Example B.0.1.** *Considering a Change-of-Order mapping* $f : \mathsf{Mat}_3(\mathbb{R}) \to \mathsf{Mat}_3(\mathbb{R})$*, obtained by reordering the standard basis* $\{e_1, e_2, e_3\}$ *to the basis* $\{e_3, e_2, e_1\}$*. For a given matrix* $A$*, we get the matrix* $f(A)$ *as follows:*

$$A \longmapsto f(A)$$

$$
\begin{array}{c}
\begin{array}{ccc} e_1 & e_2 & e_3 \end{array} \\
\begin{array}{c} e_1 \\ e_2 \\ e_3 \end{array}
\begin{pmatrix}
a_{11} & a_{12} & a_{13} \\
a_{21} & a_{22} & a_{23} \\
a_{31} & a_{32} & a_{33}
\end{pmatrix}
\end{array}
\xrightarrow{f:e_1 \leftrightarrow e_3}
\begin{array}{c}
\begin{array}{ccc} e_3 & e_2 & e_1 \end{array} \\
\begin{array}{c} e_3 \\ e_2 \\ e_1 \end{array}
\begin{pmatrix}
a_{33} & a_{32} & a_{31} \\
a_{23} & a_{22} & a_{21} \\
a_{13} & a_{12} & a_{11}
\end{pmatrix}
\end{array}
$$

**Example B.0.2.** *The graphs in Figure 3 are not distinguishable by MPNN Sato (2020) because they are locally the same. Applying* $\mathsf{SNN}_o(\alpha, (1,1))$*, a level of version* $\alpha$ *of* $\mathsf{SNN}$ *that is slightly more*

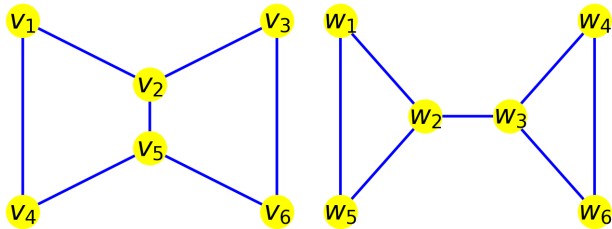

Figure 3: The graph $G$, the left one, and $H$, the right one, are not distinguishable by MPNN

*potent than MPNN, we get the following symmetric matrices* $X$ *and* $Y$ *for* $G$ *and* $H$ *respectively as the outputs of the model for these graphs.*

$$
X = \begin{pmatrix}
2 & 2 & 1 & 2 & 2 & 0 \\
2 & 3 & 2 & 2 & 2 & 2 \\
1 & 2 & 2 & 0 & 2 & 2 \\
2 & 2 & 0 & 2 & 2 & 1 \\
2 & 2 & 2 & 2 & 3 & 2 \\
0 & 2 & 2 & 1 & 2 & 2
\end{pmatrix}
\quad
Y = \begin{pmatrix}
2 & 3 & 1 & 0 & 3 & 0 \\
3 & 3 & 2 & 1 & 3 & 1 \\
1 & 2 & 3 & 3 & 1 & 3 \\
0 & 1 & 3 & 2 & 0 & 3 \\
3 & 3 & 1 & 0 & 2 & 0 \\
0 & 1 & 3 & 3 & 0 & 2
\end{pmatrix}
$$

*The entry* $ij$ *in these matrices corresponds to the count of paths between nodes* $v_i$ *and* $v_j$ *in* $\mathsf{CoSieve}(v_i, 1) \bullet \mathsf{Sieve}(v_j, 1)$ *and* $w_i$ *and* $w_j$ *in* $\mathsf{CoSieve}(w_i, 1) \bullet \mathsf{Sieve}(w_j, 1)$*. The disparity between these matrices highlights the differences between the graphs. This dissimilarity becomes more apparent when applying the set function* $\mathsf{Var}$*, while* $\mathsf{Sum}$ *and* $\mathsf{Mean}$ *yield identical values. When* $\mathsf{SNN}_o(\alpha, (1,2))$*, a more complex level of* $\mathsf{SNN}$*, is applied, we obtain the following nonsymmetric matrices, denoted as* $Z$ *and* $W$*, for graphs* $G$ *and* $H$*. Applying all three set functions results in distinct outputs, further emphasizing the dissimilarity between the graphs.*

$$
Z = \begin{pmatrix}
2 & 4 & 2 & 4 & 4 & 3 \\
5 & 3 & 5 & 4 & 6 & 4 \\
2 & 4 & 2 & 3 & 4 & 4 \\
4 & 4 & 3 & 2 & 4 & 2 \\
4 & 6 & 4 & 5 & 3 & 5 \\
3 & 4 & 4 & 2 & 4 & 2
\end{pmatrix}
\quad
W = \begin{pmatrix}
2 & 3 & 3 & 1 & 3 & 1 \\
4 & 3 & 4 & 2 & 4 & 2 \\
2 & 4 & 3 & 4 & 2 & 4 \\
1 & 3 & 3 & 2 & 1 & 3 \\
3 & 3 & 3 & 1 & 2 & 1 \\
1 & 3 & 3 & 3 & 1 & 2
\end{pmatrix}
$$

## C    Proof of Theorems

### C.1    Proof of Proposition 2.1.1

*Proof.* Let $v_i \leq_D v_j$ and $v_j \leq_D v_k$, so there are paths in $D$ from $v_i$ to $v_j$ and $v_j$ to $v_k$; hence the concatenation of these paths is a path in $D$ from $v_i$ to $v_k$ and then $v_i \leq_D v_k$.    □

### C.2    Proof of Theorem 2.1.1

*Proof.* Since Rep is surjective, it suffices to demonstrate that Rep is also injective, meaning that if $\text{Rep}(D) = \text{Rep}(D')$, then $D = D'$. According to the matrix representation definition, $\leq_D = \leq_{D'}$. For an edge $v_i \xrightarrow{e} v_j$ in $D$, it implies $v_i \leq_D v_j$, and consequently, $v_i \leq_{D'} v_j$. Suppose $v_i \xrightarrow{e} v_j$ is not a directed edge in $D'$. In that case, there must be a path in $D'$ traversing a node $v_k$ different from $v_i$ and $v_j$. This implies $v_i \leq_{D'} v_k$ and $v_k \leq_{D'} v_j$, and consequently, $v_i \leq_D v_k$ and $v_k \leq_D v_j$. Thus, there is a path in $D$ from $v_i$ to $v_j$ traversing $v_k$. However, this path is distinct from $v_i \xrightarrow{e} v_j$, contradicting the definition of directed subgraphs. Therefore, $e$ is a directed edge in $D'$. Similarly, we can demonstrate that every edge in $D'$ also belongs to $D$ with the same direction. Thus, $D = D'$.    □

### C.3    Proof of Theorem 2.2.1

*Proof.* The empty graph is its identity element, and the associativity of $\bullet$ comes from the associativity of the composition of paths. The non-commutativity is explained in Example 2.2.1.    □

### C.4    Proof of Theorem 2.2.2

*Proof.* Since directed subgraphs, together with the operation $\bullet$ generate the monoid $\text{Mod}(G)$, we just need to show that every directed subgraph can be formed by its directed edges using the operation $\bullet$. We will prove this by induction based on the number of edges. Let $D$ be a directed subgraph of $G$. There is nothing to prove if $D$ has just one directed edge. Suppose the number of edges in $D$ is $m$, and the statement is true for every directed subgraph with edges less than $m$; Our task is to show that the statement holds for $D$ as well.

Let $V_D$ be the set of nodes of $D$. According to Theorem 2.1.1, $(V_D, \leq_D)$ can be seen as a partially ordered set, implying the existence of maximal elements. A node is considered maximal if it is not the starting point of any path. Now, let $v$ be a maximal node; we choose a directed edge $w \xrightarrow{e} v$ in $D$ and remove it. The following three situations may occur:

1) producing one directed subgraph $D'$: $D$ and $D' \bigoplus e$ have the same directed edges. Since $v$ is maximal, the paths of $D$ that pass $e$ have this directed edge as their terminal edge. Then

$$\text{Paths}(D) = \text{Paths}(D') \star e$$

This follows $D = D' \bullet e$. Based on the assumption, $D'$ can be created by its edges. Then, the statement is true for $D$.

2) producing two components where one of them is an isolated node, and the other one is a directed subgraph $D'$: in this case, we first remove the isolated node and then, similar to the first case, we conclude that the statement is true for $D$.

3) producing two directed subgraphs $D'$ and $D''$ where $w \in D'$ and $v \in D''$: obviously $D$ and $D' \bigoplus e \bigoplus D''$ have the same directed edges. With an argument similar to the first part, the maximality of $v$ implies

$$\text{Paths}(D) = \text{Paths}(D') \star \{e\} \star \text{Paths}(D'')$$

and then $D = D' \bullet e \bullet D''$. Now, by the assumption that $D'$ and $D''$ can be created by their edges, the statement is true for $D$.

□

## C.5 PROOF OF THEOREM 2.3.1

*Proof.* Since the summation and multiplication of matrices are associative, the operation $\circ$ is associative. The zero matrix is the identity element of $\mathsf{Mat}_n(\mathbb{R})$ with respect to $\circ$. $\square$

## C.6 PROOF OF THEOREM 2.3.2

*Proof.* We prove the statement by induction on $k$. For $k = 2$, there is nothing to prove, which is clear from the definition. Let the statement be true for $k$; We will show it is true for $k + 1$. The associativity of $\circ$ and the induction hypothesis imply:

$$A_1 \circ A_2 \circ \cdots \circ A_k \circ A_{k+1} = (A_1 \circ A_2 \circ \cdots \circ A_k) \circ A_{k+1} =$$

$$(A_1 \circ A_2 \circ \cdots \circ A_k) + A_{k+1} + (A_1 \circ A_2 \circ \cdots \circ A_k)A_{k+1} =$$

$$\sum_{i=1}^{k} A_i + \cdots + \sum_{\sigma \in O(k,j)} A_{\sigma_1} \cdots A_{\sigma_j} + \cdots + A_1 A_2 \cdots A_k +$$

$$A_{k+1} +$$

$$(\sum_{i=1}^{k} A_i + \cdots + \sum_{\sigma \in O(k,j)} A_{\sigma_1} \cdots A_{\sigma_j} + \cdots + A_1 \cdots A_k)A_{k+1}$$

$$= \sum_{i=1}^{k+1} A_i + (\sum_{i=1}^{k} A_i A_{k+1} + \sum_{\sigma \in O(k,2)} A_{\sigma_1} A_{\sigma_2}) + \cdots +$$

$$(\sum_{\sigma \in O(k,j-1)} A_{\sigma_1} \cdots A_{\sigma_{j-1}} A_{k+1} + \sum_{\sigma \in O(k,j)} A_{\sigma_1} \cdots A_{\sigma_j}) +$$

$$\cdots + A_1 \cdots A_k A_{k+1} =$$

$$\sum_{i=1}^{k+1} A_i + \sum_{\sigma \in O(k+1,2)} A_{\sigma_1} A_{\sigma_2} + \cdots + \sum_{\sigma \in O(k+1,j)} A_{\sigma_1} \cdots A_{\sigma_j} +$$

$$\cdots + A_1 A_2 \cdots A_k A_{k+1}$$

Therefore the statement is true for $k + 1$. $\square$

## C.7 PROOF OF THEOREM 2.3.3

*Proof.* Considering that $S = \mathsf{Paths}(D_1) \star \cdots \star \mathsf{Paths}(D_k)$, let $p = p_0 p_1 \cdots p_m \in S$ be a path from $v_i$ to $v_j$ that is obtained by composition of subpaths $p_0 \in \mathsf{Paths}(D_{i_0}), \cdots, p_m \in \mathsf{Paths}(D_{i_m})$ and $1 \leq i_0 \lneqq \cdots \lneqq i_m \leq k$. The number of all such paths from $v_i$ to $v_j$ equals the $ij$ entry of the matrix $(A_{i_0} \cdots A_{i_m})$ that is a summand of $A$ as explained in Theorem 2.3.2. So the number of all paths from $v_i$ to $v_j$ in $S$ equals the $ij$ entry of $A$. Therefore, the definition of $\mathsf{Tr}$ just depends on $S$ and is independent of the choice of $D_i$s. Then $\mathsf{Tr}$ is well-defined. Based on the definition, $\mathsf{Tr}$ is a monoidal homomorphism.

Suppose $B \in \mathsf{Mom}(G)$, then there are some matrix representations $B_1, \cdots, B_l$ in $\mathsf{MatRep}(G)$ such that $B = B_1 \circ \cdots \circ B_l$. Since $\mathsf{Rep}$ is an isomorphism, there exist some directed subgraphs $C_1, \cdots, C_l$ such that $\mathsf{Rep}(C_i) = B_i$. Now, by choosing $C = C_1 \bullet \cdots \bullet C_l$, we obtain $\mathsf{Tr}(C) = B$, establishing that $\mathsf{Tr}$ is surjective. $\square$

## C.8 PROOF OF PROPOSITION 2.4.1

*Proof.* As we explained, $f$ changes the order of rows and columns. Thus, it preserves element-wise and matrix multiplications. Since $f$ is also linear, we have

$$f(A \circ B) = f(A + B + AB)$$
$$= f(A) + f(B) + f(AB)$$
$$= f(A) + f(B) + f(A)f(B)$$
$$= f(A) \circ f(B)$$

and then $f$ preserves the operation $\circ$ and this property establishes $f$ as a monoidal isomorphism. $\square$

## C.9   PROOF OF THEOREM 2.4.1

*Proof.* Since $f$ is a change in the order, it induces bijections $\mathsf{DirSub}(f)$ and $\mathsf{MatRep}(f)$ such that Diagram 4 commutes.

$$
\begin{array}{ccc}
\mathsf{DirSub}(G) & \xrightarrow{\ \mathsf{Rep}\ } & \mathsf{MatRep}(G) \\
\scriptstyle{\mathsf{DirSub}(f)}\downarrow & & \downarrow\scriptstyle{\mathsf{MatRep}(f)} \\
\mathsf{DirSub}(H) & \xrightarrow[\ \mathsf{Rep}\ ]{} & \mathsf{MatRep}(H)
\end{array}
\tag{4}
$$

Also, $f$ induces monoidal isomorphism $\mathsf{SMult}(f) : \mathsf{SMult}(G) \to \mathsf{SMult}(H)$ that sends $(M, S) \mapsto (f(M), f(S))$. According to the commutativity of the squares in Diagram 5, isomorphisms $\mathsf{Mod}(f) : \mathsf{Mod}(G) \to \mathsf{Mod}(H)$ and $\mathsf{Mom}(f) : \mathsf{Mom}(G) \to \mathsf{Mom}(H)$ can be obtained by restricting $\mathsf{SMult}(f)$ to $\mathsf{Mod}(G)$ and $\mathsf{CO}(f)$ to $\mathsf{Mom}(G)$.

$$
\begin{array}{ccccccc}
\mathsf{DirSub}(G) & \xrightarrow{\mathsf{DirSub}(f)} & \mathsf{DirSub}(H) & & \mathsf{MatRep}(G) & \xrightarrow{\mathsf{MatRep}(f)} & \mathsf{MatRep}(H) \\
\downarrow & & \downarrow & & \downarrow & & \downarrow \\
\mathsf{SMult}(G) & \xrightarrow[\mathsf{SMult}(f)]{} & \mathsf{SMult}(H) & & \mathsf{Mat}_{|V_G|}(\mathbb{R}) & \xrightarrow[\mathsf{CO}(f)]{} & \mathsf{Mat}_{|V_H|}(\mathbb{R})
\end{array}
\tag{5}
$$

The commutativity of the right square in Diagram 1 directly follows from the definition of $\mathsf{Mom}(f)$. As illustrated in Diagram 4, the left square in Diagram 1 is shown to be commutative for the generators of monoids, establishing the commutativity of this square. $\qquad\square$

## C.10   PROOF OF THEOREM 2.4.2

*Proof.* We begin by demonstrating that $f$ establishes a one-to-one correspondence between the edges of $G$ and $H$. It is evident that a matrix with a single non-zero entry in either $\mathsf{Mom}(G)$ or $\mathsf{Mom}(H)$ corresponds to a matrix transformation of an element in $\mathsf{Mod}(G)$ or $\mathsf{Mod}(H)$, respectively, each representing a single directed edge.

For an edge $v_i \,\text{---}\, v_j$ in $G$, let $e$ be the directed edge $v_i \to v_j \in \mathsf{Mod}(G)$; then $A = \mathsf{Tr}_G(e)$ has one non-zero entry, and since $f$ is a linear isomorphism, $f(A)$ has one non-zero entry, and, based on the assumption, it belongs to $\mathsf{Mom}(H)$. So $f(A)$ is a matrix transformation of a directed edge $c : u_k \to u_l$ in $\mathsf{Mod}(H)$. Similarly, let $B \in \mathsf{Mom}(G)$ be the matrix transformation of $e' : v_j \to v_i$ and then $f(B) \in \mathsf{Mom}(H)$ is a matrix transformation of some directed edge $c' : u_{l'} \to u_{k'}$ in $\mathsf{Mod}(H)$. Since $e$ can be followed by $e'$, $e \bullet e'$ has three paths. This implies $\mathsf{Tr}_G(e \bullet e')$ has three non-zero entries. On the other hand, $\mathsf{Tr}_G(e \bullet e') = \mathsf{Tr}_G(e) \circ \mathsf{Tr}_G(e') = A \circ B = A + B + AB$; then $AB \neq 0$ and consequently $f(A)f(B) = f(AB) \neq 0$. The equation

$$
\begin{aligned}
\mathsf{Tr}_H(c \bullet c') &= \mathsf{Tr}_H(c) \circ \mathsf{Tr}_H(c') \\
&= f(A) \circ f(B) \\
&= f(A) + f(B) + f(A)f(B)
\end{aligned}
$$

says that the matrix transformation corresponding to $c \bullet c'$ has three non-zero entries and so $c \bullet c'$ contains three paths. Then $c$ must be followed by $c'$ and this yields $u_l = u_{l'}$. Similarly, $u_k = u_{k'}$ can be shown. Therefore, $f$ gives a one-to-one mapping between the edges of $G$ and $H$.

To prove the correspondence between the nodes of two graphs, let $v_x$ be a node in $G$, connected to $v_i$ in which $j \neq x$ and $C$ and $f(C)$ be the matrix transformations of $a : v_i \to v_x \in \mathsf{Mod}(G)$ and $b : u_y \to u_z \in \mathsf{Mod}(H)$, respectively. Since $e'$ is followed by $a$ in $\mathsf{Mod}(G)$, with the same reasoning as above, $c'$ must be followed by $b$ in $\mathsf{Mod}(H)$ and this means $u_k = u_y$. So $f$ also gives a one-to-one mapping between nodes of graphs compatible with edges. Then, $G$ and $H$ are isomorphic. $\qquad\square$

## C.11 Proof of Theorem 2.5.1

The role of neighborhoods in MPNN is like a sink such that messages move to the center of the sink. For a node $v_k$ with neighborhood $N_k$ containing $v_{k_1}, v_{k_2}, \cdots, v_{k_m}$, we depict this sink in Figure 4 by denoting directed edge from $v_{k_i}$ to $v_k$ by $e_i : v_{k_i} \to v_k$. This sink can be considered as a directed

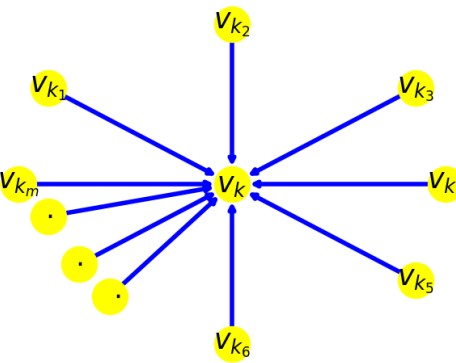

Figure 4: Visualizing a neighborhood by representing it as a directed subgraph

subgraph. As an element of $\mathsf{Mod}(G)$, it can be represented as follows:

$$S_k = e_1 \bullet e_2 \bullet \cdots \bullet e_m$$

Since the directed edges $e_i$ and $e_j$ appearing in $S_k$ are not composable, we observe $e_i \bullet e_j = e_j \bullet e_i$, rendering the order in $S_k$ unimportant. The cover obtained by $S_k$s is exactly the cover of the neighborhoods. Let $T_k = \mathsf{Tr}(S_k)$ and $A_i = \mathsf{Tr}(e_i)$. Thus $A_i$ has 1 in the entry $k_i k$ and 0 for all other entries. The matrix transformation of $e_i \bullet e_j$ has just two non-zero entries and $\mathsf{Tr}(e_i \bullet e_j) = A_i + A_j + A_i A_j$. Then $A_i A_j = 0$ for $1 \le i \le m$ and $1 \le j \le m$. Theorem 2.3.2 implies

$$T_k = \mathsf{Tr}(S_k) = A_1 \circ A_2 \circ \cdots \circ A_m$$
$$= A_1 + A_2 + \cdots + A_m$$

As a result, the column $k$ of $T_k$ aligns with the column $k$ of the adjacency matrix of graph $G$, while the remaining columns are filled with zeros. Transforming the cover $\{S_k\}$ yields a collection of $|V|$ matrices, each containing a single column from the adjacency matrix. In the GGNN framework, summation is an allowed operation, enabling the construction of the adjacency matrix by performing the summation on this matrix collection. Hence, neighborhoods can function as a cover within the framework of GGNN, with the adjacency matrix serving as an interpretation of this cover.

## C.12 Proof of Theorem 3.1.1

*Proof.* Since the definition of sets $M_i(v)$s is based on the neighborhoods, for a graph isomorphism $f : G \to H$, $f(M_i(v)) = M_i(f(v))$. This follows $\mathsf{Mod}(f)(D_i(v)) = D_i(f(v))$. Since $\mathsf{Mod}(f)$ is a monoidal homomorphism, we get:

$$\mathsf{Mod}(f)(\mathsf{Sieve}(v,k)) = \mathsf{Mod}(f)(D_k(v) \bullet \cdots \bullet D_0(v))$$
$$= \mathsf{Mod}(f)(D_k(v)) \bullet \cdots \bullet \mathsf{Mod}(f)(D_0(v))$$
$$= D_k(f(v)) \bullet \cdots \bullet D_0(f(v))$$
$$= \mathsf{Sieve}(f(v),k)$$

Based on Theorem 2.4.1, $\mathsf{Mom}(f)(\mathsf{Image}(v,k)) = \mathsf{Image}(f(v),k)$. Also, $\mathsf{CO}(f)$ preserves the rest of the computations in the algorithm, so SNN is invariant. $\qquad\square$

## D Explanation for constructing a model in GGNN framework

The process of designing a GNN model within this framework is outlined as follows:

1) For a given graph $G$, the process involves selecting a collection $\mathcal{C}_G$ of elements from $\mathsf{Mod}(G)$ to serve as a cover for $G$. These elements can be generated using $\mathsf{DirSub}(G)$ and the binary operation $\bullet$. Notably, Theorem 2.2.2 ensures the ability to create any suitable and desired elements by leveraging directed edges and the operator $\bullet$.

2) Next, the chosen cover is transformed into a collection of matrices within $\mathsf{Mom}(G)$. During this transformation, the operation $\circ$ and other elements of $\mathsf{Mom}(G)$ can be employed to convert the original collection into a new one. The resulting output at this stage is denoted by $\mathcal{A}_G$.

3) By utilizing $\iota$, the collection obtained in the second stage transitions into a larger and more equipped space, a suitable environment for enrichment. This stage leverages all the operations outlined in Proposition 2.4.1 to complete the model's design. Following the processing of $\mathcal{A}_G$ in this stage, we obtain a new collection of matrices denoted by $\mathcal{M}_G$, representing the model's output.

Hence, a model is a mapping that associates a collection of matrices $\mathcal{M}_G$ with a given graph $G$. $\mathcal{M}_G$ plays a role akin to the adjacency matrix and provides an interpretation of the chosen cover for use in various forms of message passing. While the second and third stages can be merged, we prefer to emphasize the significance of $\mathsf{Tr}$ in this process.

This construction of a model is appropriate for tasks such as node classification. For graph classification, we need an invariant construction. Based on Theorem 2.4.1, a graph isomorphism $f : G \to H$ transform the triple $(\mathcal{C}_G, \mathcal{A}_G, \mathcal{M}_G)$ to a triple $(\mathcal{C}'_H, \mathcal{A}'_H, \mathcal{M}'_H)$ for graph $H$ and this may be different from $(\mathcal{C}_H, \mathcal{A}_H, \mathcal{M}_H)$. So a model constructed in the GGNN framework is invariant if for every graph isomorphism $f : G \to H$, the maps $\mathsf{Mod}(f)$, $\mathsf{Mom}(f)$ and $\mathsf{CO}(f)$ induce one-to-one correspondences between $\mathcal{C}_G$ and $\mathcal{C}_H$, $\mathcal{A}_G$ and $\mathcal{A}_H$, and $\mathcal{M}_G$ and $\mathcal{M}_H$, respectively. The model SNN is an example of an invariant model.

