# OpenReview forum: "Grothendieck Graph Neural Networks Framework: An Algebraic Platform for Crafting Topology-Aware GNNs"
_ICLR.cc/2025/Conference — ICLR 2025 Conference Withdrawn Submission_

### Official Review · Reviewer_REnp · 2024-10-31

**Soundness:** 3
**Presentation:** 2
**Contribution:** 3
**Rating:** 6
**Confidence:** 3

**Summary:**

This paper presents the Grothendieck Graph Neural Networks (GGNN) framework, an algebraic approach for constructing topology-aware Graph Neural Networks by generalizing neighborhood structures through algebraic covers. The GGNN framework introduces a novel approach to define high-order adjacency via directed subgraph and graph cover. Based on this, the authors propose the Sieve Neural Networks (SNN) model, leveraging the sieve concept from category theory to perform hihg-order message passing based on the derived matrix notation.

**Strengths:**

1. The GGNN framework creatively applies algebraic topology, offering a new paradigm for designing message-passing strategies that extend beyond conventional neighborhood-based approaches.

2. By using algebraic covers, GGNN can capture higher-order relationships in graphs, which enhances GNN expressiveness and aligns well with the intrinsic structure of graph data.

3. SNN demonstrates strong performance in experiments, particularly in distinguishing structurally complex graphs.

**Weaknesses:**

1. While GGNN introduces a novel approach for incorporating higher-order graph structural information into the message-passing framework, it does not include comparisons with other related methods, such as path-based GNNs[1-3] or substructure-based GNNs[4-5], and $K$-hop message passing GNNs[6-7], which also integrate higher-order structural information. It would strengthen the work to explain GGNN’s specific advantages over these methods, particularly concerning its expressive power.

2. The readability of the paper could be enhanced. For instance, including pseudo-code would provide a clearer procedural understanding of the GGNN method. Additionally, a figurative example demonstrating how SNN distinguishes strongly regular graphs where the 3-WL test fails would be beneficial for clarity.

3. Since the framework relies on concepts from category theory, it would benefit the readers if some background knowledge on this topic were provided to make the paper more self-contained.

4. The experimental validation is primarily limited to TUDatasets. Including other datasets commonly used for GNN expressivity, such as OGB or QM9, would help assess the generalizability of GGNN across varied graph types.

5. The results on the SR and CSL datasets are not shown in a tabular format.


__References__

1. Gaspard Michel, et al., Path Neural Networks: Expressive and Accurate Graph Neural Networks. ICML2023
2. Ralph Abboud, et al., Shortest Path Networks for Graph Property Prediction, LoG2022
3. Kong, et al., Geodesic graph neural network for efficient graph representation learning. NeurIPS2022
4. Zhang, et al., Nested Graph Neural Networks, NeurIPS2021
5. Frasca, et al., Understanding and Extending Subgraph GNNs by Rethinking Their Symmetries, NeurIPS2022
6. Feng, et al., How Powerful are K-hop Message Passing Graph Neural Networks, NeurIPS2022
7. Yao et al., Improving the Expressiveness of K-hop Message-Passing GNNs by Injecting Contextualized Substructure Information, KDD2023

**Questions:**

1. What is the advantages of GGNN and SNN over the existing methods in enhancing the expressive power of GNN in terms of distinguishing graph structures?

---

> ### Author Response · Authors · 2024-11-17
> **Reply to Reviewer REnp**
>
> Thank you for your valuable feedback. Below, we address your comments and questions in detail.
>
> $\textbf{Responses to Weaknesses}$
>
> $\textbf{1. Comparisons with Related Methods}$
>
> We appreciate your suggestion to compare GGNN with other methods, such as path-based GNNs, substructure-based GNNs, and $k$-hop message-passing GNNs. These methods also integrate higher-order structural information, but GGNN distinguishes itself by providing a fundamentally new perspective through its algebraic framework. Specifically:
>
> $\textbf{Novel Representations for Graphs:}$ While other methods rely on the adjacency matrix as the foundational representation of graphs, GGNN provides infinitely many alternatives to the adjacency matrix, thereby expanding the design space for GNN models.
>
> $\textbf{Algebraic Tools:}$ The framework introduces algebraic structures such as the monoid $\mathsf{Mod}(G)$ and the binary operation $\bullet$, which allows defining alternative concept for neighborhoods called covers. These covers serve as flexible alternatives to traditional neighborhoods.
>
> $\textbf{Expressive Power:}$ Theorems 2.4.1 and 2.4.2 in the paper prove that GGNN offers a $\textit{unique up-to-isomorphism algebraic description}$ for graphs. As a result, GGNN can always distinguish between non-isomorphic graphs by choosing appropriate covers, offering a theoretical guarantee not provided by many existing methods.
>
>
> $\textbf{2. Readability Improvements}$
>
> We acknowledge the need for improved clarity in presenting the framework. To enhance readability, we will:
>
> Add a $\textbf{figurative example}$ illustrating how $\mathsf{SNN}$ distinguishes strongly regular graphs, specifically those that the 3-WL test fails to differentiate.
>
> Incorporate background information on $\textbf{category theory}$ in the preliminaries to make the paper more self-contained and accessible to readers unfamiliar with this topic.
>
> Present the results on the SR and CSL datasets in a clear $\textbf{tabular format}$.
>
> $\textbf{Response to Questions}$
>
> $\textbf{Advantages of GGNN:}$
>
> GGNN introduces a fundamentally new way of viewing graph neighborhoods by leveraging algebraic structures:
>
> The monoid $\mathsf{Mod}(G)$ provides infinitely many alternatives to traditional neighborhoods, allowing researchers to consider graphs from diverse perspectives tailored to specific goals.
>
> GGNN introduces the monoidal homomorphism $\mathsf{Tr}$, which transforms covers into collections of matrices as alternatives to the adjacency matrix. These diverse numerical representations can be used as input for other GNN models.
>
> Theorems 2.4.1 and 2.4.2 prove that GGNN gives a $\textit{unique up-to-isomorphism description}$ of a graph, enabling it to distinguish any two non-isomorphic graphs.
>
>
> $\textbf{Advantages of}$ $\mathsf{SNN}$:
>
> $\mathsf{SNN}$ exemplifies how GGNN can be applied to design a topology-aware GNN model:
>
> $\mathsf{SNN}$ is specifically designed to extract deep topological features of graphs. For example, in Example B.0.2, it distinguishes between holes formed by 3 edges and those formed by 4 edges.
>
> Experiments demonstrate $\mathsf{SNN}$'s ability to differentiate strongly regular graphs, achieving $100$ percent accuracy on these datasets. This highlights its capacity to handle one of the most challenging graph isomorphism tasks.
>
> $\mathsf{SNN}$ resists oversmoothing by considering all paths of specific lengths between nodes during message passing. As a result, even distant nodes can exchange information effectively.
>
>
> We hope this response addresses your concerns and provides a clearer understanding of the contributions and strengths of GGNN and $\mathsf{SNN}$.

---

> ### Comment · Reviewer_REnp · 2024-11-23
>
> Thank you to the authors for their detailed responses. After carefully reviewing the reply, my concerns are as follows:
>
> - I appreciate the authors' clarification regarding the expressive power of GGNN (up to isomorphism in algebraic description). However, **how GGNN directly compares with other similar methods that leverage higher-order structural information remains unclear. As no formal theoretical analysis is provided on this point and these methods are not included as baselines**, my concern persists.
>
> - Another concern is the time complexity of the proposed method. This raises further concerns about its **efficiency**.
>
> - Lastly, while I thank the authors' acknowledgement to improve the readability of the paper, the required revisions appear to be non-trivial.
>
> Based on these points, while I recognize the contributions of this work, I can't raise my score for now.

---

> > ### Author Response · Authors · 2024-11-23
> >
> > $\textbf{Response to Reviewer Comments}$
> >
> > We sincerely thank the reviewer for their valuable feedback.
> >
> > $\textbf{Comparing GGNN with Higher-Order GNNs}$
> >
> > $\textbf{1.}$ While other higher-order GNNs represent specific GNN models, GGNN is not merely a model but a $\textbf{framework}$. It allows researchers to view a graph from various perspectives by defining alternatives to neighborhoods, thereby providing a platform to create a broad range of GNNs.
> >
> > $\textbf{2.}$ As mentioned in Line 44, while other higher-order GNNs focus on analyzing specific patterns or subgraphs, GGNN enables users to define alternatives for neighborhoods more precisely, rather than relying on predefined patterns.
> >
> > $\textbf{3.}$ Unlike higher-order GNNs, which are constrained by their suggested patterns, GGNN offers $\textbf{infinitely many alternatives}$ for neighborhoods. These alternatives naturally extend the concept of neighborhoods, making the framework significantly more flexible and expressive.
> >
> > $\textbf{Time Complexity}$
> >
> > $\textbf{1.}$ It is important to emphasize that GGNN itself does not have a defined time complexity because it is a $\textbf{framework}$ rather than a specific model. We have already introduced four models based on GGNN:
> >
> > $\textbf{a.}$ MPNN (Theorem 2.5.1),
> >
> > $\textbf{b.}$ $\mathsf{SNN}$,
> >
> > $\textbf{c.}$ Two additional models presented in response to Reviewer kNJA's comments.
> >
> > $\textbf{Question:}$ Which model's time complexity should be considered the time complexity of GGNN?
> >
> > $\textbf{Answer:}$ None of them. GGNN is a framework and not tied to the time complexity of any specific implementation.
> >
> > $\textbf{2.}$ As stated in Line 463, the time complexity of $\mathsf{SNN}$ is $\mathcal{O}(|E| \cdot |V|^2)$, which is comparable to other higher-order GNNs.
> >
> > To the best of our knowledge, $\mathsf{SNN}$ is the only GNN model capable of distinguishing between all graphs in the collection of strongly regular graphs. Furthermore, as noted in our earlier response to Reviewer kNJA, $\mathsf{SNN}$ achieved the best results on the BREC benchmark. Considering these capabilities, we believe this time complexity is reasonable and justified.
> >
> > We once again thank the reviewer for their feedback.

---

> > > ### Comment · Reviewer_REnp · 2024-11-25
> > >
> > > Dear authors, thank you for your reply. My concerns regarding the time complexity and novelty have been addressed, and I have raised my score to 6.
> > >
> > > However, it seems the paper includes significant revisions due to:
> > > - the addition of discussions and formal analyses comparing with other methods that incorporate higher-order structural information;
> > > - the inclusion of experiments on relevant baselines;
> > > - improvements to the clarity of the background knowledge and the method;

---

> > > > ### Author Response · Authors · 2024-11-25
> > > >
> > > > Dear Reviewer
> > > >
> > > > Thank you for reconsidering our responses and raising your score. We sincerely appreciate your constructive feedback.
> > > >
> > > > Best Regards

---

### Official Review · Reviewer_dtWY · 2024-11-01

**Soundness:** 2
**Presentation:** 1
**Contribution:** 2
**Rating:** 3
**Confidence:** 3

**Summary:**

This paper presents the Grothendieck Graph Neural Network framework, a approach that enhances GNNs by generalizing neighborhood structures using algebraic covers. Utilizing category theory and Grothendieck topologies, GGNN redefines graph covers, which are then converted into matrix representations to expand GNN architectural possibilities. The authors introduce the Sieve Neural Network within this framework, leveraging category theory to create sieve-based covers that support richer message-passing strategies. This model shows improved emperical performance.

**Strengths:**

This paper tries to design a new type of GNNs from the perspective of category theory. This is a novel and promising research direction. Empirical studies show improved performance of the new GNN.

**Weaknesses:**

The major weakness is perhaps the writing style and structure of this paper. I read the paper three times but still could not grasp what benefit the new GNN framework provides. In particular, the motivation/intuition is burried deep in lots of definitions, would be better to mention it clearly and early.
The construction of the framework is very complex and confusing and involves a lot of new terms and concepts that are not well explained. The new GNN has a very high complexity of $O(n^4)$ so I would expect the authors to justify this trade-off in complexity by benefits but couldn't find such discussion.
* The construction of GGNN involves a lot of newly introduced terms and concepts, but what benefit does it provide? How are the the directed subgraphs generated and how does such generatoin affect GGNN's power? Would be good to have specific sections with explanations of the benefits of GGNN and the process for generating directed subgraphs. Also a discussion on how different generation methods might impact GGNN's expressive power is better.
* The definition of directed subgraph is confusing. a gentle introduction of "acyclic" graph somewhere would be better. Adding it somewhere in preliminary would be nice.
* Thm 2.1.1 is trivial: Rep(D) is just a directed adjacency matrix. If the authors think this is important, please elaborate.
* I failed to find results of on SR and CSL datasets mentioned in Sec 3.4 except text description.
* Related work is limited and should be improved. There are many works try to enrich feature aggregation with alternative neighourhood definition e.g. [1], please consider compare with these works.
* wrong citation style used (I think author used \cite where it should be \citep), making a complex paper even harder to read

[1] Li, Shouheng, Dongwoo Kim, and Qing Wang. "Local vertex colouring graph neural networks." International Conference on Machine Learning. PMLR, 2023.

**Questions:**

* What (theoretical) superiority does GGNN gives us over other GNNs to justify the high complexity cost? Please be concrete.
* How are directed subgraphs chosen? Does the choice affect the power of GGNN?

---

> ### Author Response · Authors · 2024-11-17
> **Reply to Reviewer dtWY**
>
> Thank you for taking the time to review our paper.
>
> $\textbf{Main Contribution Clarification}$
>
> The GGNN framework provides a systematic improvement in the GNN literature by addressing a fundamental question: *Where does the concept of neighborhood originate, and does this origin allow for alternative definitions?* To explore this question, we developed an algebraic theory that naturally involves new terms and definitions. Below, we clarify the framework’s contributions.
>
> In this paper, the origin of the neighborhood concept is formalized as a monoid $\mathsf{Mod}(G)$, which contains infinitely many elements. The alternatives to neighborhoods are presented as collections of finitely many elements of this monoid, referred to as *covers*. The GGNN framework provides algebraic tools, particularly the binary operation $\bullet$, to construct diverse covers. We further proved that all elements of $\mathsf{Mod}(G)$ can be generated using directed edges and the operation $\bullet$. Additionally, the framework introduces a monoidal homomorphism $\mathsf{Tr}$, which transforms covers into collections of matrices, serving as alternatives to adjacency matrices.
>
> To demonstrate the practical utility of this framework, we introduced the $\mathsf{SNN}$ model. This example illustrates how to leverage the tools in GGNN to design a GNN model. The construction of $\mathsf{SNN}$ showcases the process of utilizing the framework to create models tailored to specific objectives.
>
> $\textbf{Responses to Specific Comments}$
>
> $\textbf{1. Directed Subgraphs and Their Role}$
>
> The definition of directed subgraphs as a basis for the monoid $\mathsf{Mod}(G)$ is carefully chosen and plays a fundamental role in the theory. While we understand the suggestion to incorporate acyclic graphs, this substitution would disrupt the theoretical foundation and is therefore not viable.
>
> $\textbf{2. Clarification on $\mathsf{Rep}(D)$}$
>
> As illustrated in Figure 1, $\mathsf{Rep}(D)$ is not equivalent to a directed adjacency matrix. Instead, it encodes critical path information within a directed subgraph, making it more expressive than a standard adjacency matrix. We will elaborate on this distinction in the revised version.
>
> $\textbf{3. Experimental Results on SR and CSL}$
>
> In the experimental section, we stated that $\mathsf{SNN}$ achieves perfect differentiation between all graphs in the strongly regular (SR) and CSL collections, resulting in an accuracy of $100$ percent. We appreciate your suggestion and will include a detailed table to present these results more clearly in the revised manuscript.
>
> $\textbf{4. Related Work}$
>
> We acknowledge the limited discussion of related work and will update this section to include additional references, such as [1].
>
> $\textbf{Answers to Reviewer Questions}$
>
> $\textbf{Q1: Theoretical Superiority and Complexity Trade-off}$
>
> GGNN is a framework rather than a standalone model. Its primary advantage lies in its ability to provide algebraic tools for designing GNN models tailored to specific tasks. For example, $\mathsf{SNN}$ demonstrates how GGNN enables the extraction of rich topological information. Its outstanding performance on strongly regular graph collections highlights this capability. Furthermore, $\mathsf{SNN}$ exemplifies how concepts from other fields (e.g., sieves from category theory) can be integrated into GNN model design via GGNN.
>
> Each model created using GGNN has its own complexity, depending on its specific design and purpose.  The complexity of a specific instance (e.g., $\mathsf{SNN}$) should not be generalized as a limitation of the framework itself. For $\mathsf{SNN}$, the time complexity for sparse graphs is $\mathcal{O}(|E| \cdot |V|^2)$. This complexity is justified by the model’s focus on extracting detailed topological information.
>
> $\textbf{Q2: Directed Subgraph Generation and GGNN Power}$
>
> According to Theorem 2.2.2, appropriate monoidal elements can be generated using directed edges and the binary operation $\bullet$. This flexibility allows researchers to define directed subgraphs or monoidal elements according to their objectives. The GGNN framework does not impose constraints on these choices, enabling it to support a wide range of use cases. For examples of models and their corresponding directed subgraphs, please refer to the examples provided in our response to Reviewer kNJA.
>
> It is also worth noting that $\mathsf{SNN}$ is used as a preprocessing tool to transform datasets by replacing adjacency matrices with its outputs. Once the transformation is complete, a chosen message-passing operator can be applied, minimizing the scalability concerns during training.
>
> We hope these clarifications address your concerns and provide a better understanding of the GGNN framework and its contributions.

---

> ### Author Response · Authors · 2024-11-26
>
> $\textbf{Reviewer Comment: What can it provide beyond other GNNs?}$
>
> This framework establishes a novel context for defining the concept of covers for graphs, enabling their transformation into matrix forms that can be integrated into the message-passing process.
>
> The concept of covers in the GGNN framework diverges from the traditional view of graphs derived from simple neighborhood definitions. By generating multiple perspectives of a graph through these covers, the framework enables the design of a variety of message-passing strategies for GNNs. These capabilities unlock new flexibility and potential for analyzing complex graph structures in ways that traditional GNNs cannot accommodate.
>
> $\textbf{Reviewer Comment: The authors have mentioned the ability to differentiate strongly regular graphs, but the result is descriptive.}$
>
> This point has already been addressed in our earlier response:
> In the experimental section, we explicitly demonstrated that $\mathsf{SNN}$ achieves perfect differentiation among all graphs in the strongly regular (SR) and CSL collections, attaining an accuracy of $100$ percent.
> The results are not merely descriptive but are backed by rigorous experimentation.
>
> $\textbf{Reviewer Comment: The authors provide further explanation of some definitions in the response, e.g., Rep(D), but the description does not match the definition in the paper.}$ $\textbf{ I feel a major revision is needed to refine these definitions and improve the readability of this paper.}$
>
> The explanation provided in the response aligns entirely with the definitions presented in the paper. The framework is built on a well-organized and rigorous theoretical foundation with precise definitions, as documented in the manuscript.
>
> We respectfully urge the reviewer to revisit the definitions in the paper carefully. Misinterpretations or oversights in reviewing can lead to unnecessary criticisms, and we believe a fair assessment of our work requires close attention to the provided details.

---

> > ### Comment · Reviewer_dtWY · 2024-11-28
> >
> > Thanks for your reply.
> > To sufficiently address my concerns regarding strongly regular graphs, I would expect the authors to provide a theoretical lemma and proof to show why your model can distinguish them. As I believe this is one of the main advantages of SSN and it could also partially address my concerns regarding the benefit beyond other GNNs.
> >
> > The authors suggest the design can "extract rich topological strucutre" and "these capabilities unlock new flexibility...that traditional GNNs cannot accommodate." This is the main contribution but the message is not concrete: What topological strucutre exactly can this model extract and traditinoal GNNs cannot? I suggest authors to give concrete theoretical analysis and examples to support this argument.
> >
> > For my concerns regarding definition and terminologies, the authors mentioned they will elaborate them in the revision, including figures, but I failed to find any changes so far.
> > Last I hope the authors can fix the cite/citep issues to making the reading a little easier for me.

---

### Official Review · Reviewer_wDej · 2024-11-02

**Soundness:** 2
**Presentation:** 2
**Contribution:** 2
**Rating:** 3
**Confidence:** 3

**Summary:**

This paper introduces a framework that extends message-passing graph neural networks through generalizing the definition of neighborhood via algebraic representation of the covers of graphs. With the covers mapping into matrix representations, the framework is able to extend GNNs into new architectures, e.g., sieve neural networks, by leveraging relevant theories. Experiments are performed on datasets to verify the efficacy of the proposed approach.

**Strengths:**

1. The proposed framework seems to be a new concept inspired by generalizing conventional message passing with algebraic covers.

2. The algorithmic analysis on the method is detailed.

**Weaknesses:**

1. The complexity of the proposed approach is O(n^4) which significantly limits the scalability towards larger graphs, see Q1.

2. The experiment results are preliminary with part of those conveying vague information and part of those lacking important baselines, see Q2 and Q3.

3. Missing discussions with more recent literatures, see Q4. The related works mentioned in this paper are earlier than 2023 with many missing.

Minor:
There are numerous inappropriate usages of citet and citep throughout the paper which hinders the readability.


To me the paper might require significant efforts in improving the presentation and addressing the concerns in inadequate experimental evaluation and insufficient discussions and reference to related works.

**Questions:**

1. It would be helpful to provide the runtime of different approaches to give an idea on the scalability of the approach. In particular, runtime on different scales of the graphs would be appreciated. Since the model is of complexity O(n^4), I believe this is already a limitation which is even higher than some of the recent advanced approaches.

2. The SR (line 474) and CSL (line 485) of the experiments only convey vague information with no quantitative results presented.

3. The results in Table 1 are not quite convincing since the strong baselines are not included, such as [1]. The work has been cited and discussed in related works but not referred to in Table 1.

4. Missing related works, such as [2], [3], [4], [5].

[1] Feng et al. How Powerful are K-hop Message Passing Graph Neural Networks. NeurIPS'22.

[2] Zhang et al. Rethinking the Expressive Power of GNNs via Graph Biconnectivity. ICLR'23.

[3] Zhang et al. Beyond Weisfeiler-Lehman: A Quantitative Framework for GNN Expressiveness. ICLR'24.

[4] Wijesinghe et al. A New Perspective on "How Graph Neural Networks Go Beyond Weisfeiler-Lehman?". ICLR'22.

[5] Zhao et al. A Practical, Progressively-Expressive GNN. NeurIPS'22.

---

> ### Author Response · Authors · 2024-11-17
> **Reply to Reviewer wDej**
>
> Thank you for taking the time to provide valuable feedback on our work. We address your concerns below.
>
> $\textbf{Main Contribution Clarification}$
>
> As stated in the paper, our primary contribution lies in algebraically extending the concept of neighborhoods in graphs. We achieve this through the introduction of the GGNN framework, which provides a foundation for creating novel GNN models. $\mathsf{SNN}$ serves as an example to illustrate the practical application of the GGNN framework. A significant portion of the paper (approximately 13 pages) is devoted to presenting a $\textbf{Strong Algebraic Theory}$ that explores and extends the concept of neighborhoods. The models, such as $\mathsf{SNN}$ and the examples provided in response to Reviewer kNJA, demonstrate the transformative capability of the GGNN framework. Each model has its own complexity, and the complexity of a specific instance (e.g., $\mathsf{SNN}$) should not be generalized as a limitation of the framework itself.
>
> $\textbf{Responses to Specific Comments}$
>
> $\textbf{1. Scalability and Complexity}$
>
> We emphasize that most real-world graphs are sparse. As noted in the paper, for sparse graphs, the time complexity of $\mathsf{SNN}$ reduces to $\mathcal{O}(|E| \cdot |V|^2)$. Since $\mathsf{SNN}$ is specifically designed to extract topological information from graphs, this complexity is justified by the richness of the information it captures. Furthermore, it is important to note that $\mathsf{SNN}$ is primarily used to preprocess datasets by replacing the adjacency matrix with its output. Once this transformation is complete, the training phase proceeds with any chosen message-passing operator, effectively mitigating scalability concerns during training.
>
> $\textbf{2. Experiments on Strongly Regular Graphs (SR) and CSL}$
>
> In the experimental section, we demonstrated that $\mathsf{SNN}$ achieves perfect differentiation between all graphs in the strongly regular (SR) and CSL collections. This corresponds to an accuracy of $100$ percent. These results highlight the model’s ability to capture and utilize deep topological features of a graph. We will revise the text to make this point clearer and to include quantitative results for better clarity.
>
> $\textbf{3. Table 1 and Strong Baselines}$
>
> We appreciate your observation regarding Table 1. The absence of certain strong baselines, such as [1], is noted, and we will revise the table to include comparisons with these methods to provide a more comprehensive evaluation.
>
> $\textbf{4. Missing Related Works}$
>
> Thank you for pointing out the missing references, including [2], [3], [4], and [5]. We acknowledge the importance of discussing recent advancements and will update the related works section accordingly. This will ensure a broader and more current contextualization of our contributions.
>
> $\textbf{Minor Issue: Inappropriate Usage of Citations}$
>
> We regret the inappropriate usage of \texttt{citet} and \texttt{citep} throughout the paper. This will be corrected in the revised version to improve the readability and professionalism of the manuscript.
>
> We hope that these responses address your concerns and help clarify our contributions.

---

> > ### Comment · Reviewer_wDej · 2024-11-26
> >
> > Thank the authors for the rebuttal. However, I did not find any revisions regarding the missing baselines and related literature. I believe this is important and thus will keep my score.

---

### Official Review · Reviewer_kNJA · 2024-11-04

**Soundness:** 3
**Presentation:** 3
**Contribution:** 3
**Rating:** 6
**Confidence:** 3

**Summary:**

The paper introduces Grothendieck Graph Neural Networks(GGNN) framework, which systematically defines and refines diverse covers for graphs from an algebraic perspective. The framework uses matrix representations of covers and utilizes change-of-order mapping to enrich GNN operations. The authors further propose Sieve Neural Networks (SNN) based on the concept of sieves in category theory, and provide corresponding analysis. Experiments on SR and CSL datasets verifies the expressive power of SNN.

**Strengths:**

The GGNN framework can systematically define and refine diverse covers of graphs from an algebraic perspective, which is novel and offers plenty of design space. Overall, the paper is well presented. I checked with the proofs and have not found fatal mistakes.

**Weaknesses:**

* Although the GGNN framework provides a general recipe for new GNNs, the authors only present one specific new GNN (i.e., SNN). The paper could benefit from more efforts elaborating on this.

* The theoretical analysis does not involve the connection with the WL hierarchy. What is the fine-grained expressivity of the proposed methods? The paper can also further benefit from more comparison and relationship with existing works, including isomorphism/homomorphism expressivity and other topology-aware GNNs.

* The experiments can be improved. (1) SR and CSL can only partially reflect expressivity in a very coarse manner. Can you conduct experiments on BREC[1], a more fine-grained expressivity benchmark? (2) TUdataset typically consists of very small and easy datasets, which is not convincing. Experiments on more common benchmarks (e.g., ZINC and QM9) are strongly encouraged.

[1] Wang, Y., & Zhang, M. (2023). Towards better evaluation of gnn expressiveness with brec dataset. arXiv preprint arXiv:2304.07702.

**Questions:**

* Given that GGNN is a general framework, can you offer some examples of new GNN architectures other than SNN? Brief descriptions suffice.

* Can you elaborate on the expressivity compared with k-WL so that the results align with the experiments on SR and CSL?

* What is the scalability and performance of SNN in more large-scale real-world datasets?

---

> ### Author Response · Authors · 2024-11-17
> **Reply to Reviewer kNJA**
>
> Thank you for taking the time to provide insightful feedback on our work. Below, we address your concerns in detail.
>
> $\textbf{Main Contribution Clarification}$
>
> The primary goal of this paper is to algebraically extend the notion of neighborhoods in graphs, enabling greater flexibility and efficiency while maintaining simplicity in practical applications. This extension is designed to seamlessly integrate covers into the construction of GNN models. The proposed GGNN framework serves as a versatile tool for developing new GNN architectures, demonstrated by the example of $\mathsf{SNN}$.
>
> $\textbf{Responses to Specific Comments}$
>
> $\textbf{1. Examples of GNN Architectures Beyond}$ $\mathsf{SNN}$
>
> To illustrate the utility of the GGNN framework in designing novel GNN architectures, we provide two additional examples:
>
> $\textbf{First Example:}$
>
> We construct a cover different from the traditional neighborhood or the $\mathsf{SNN}$ cover by forming a collection of elements in the monoid $\mathsf{Mod}(G)$. Each node $v \in G$ is associated with the monoidal element $2\text{-}\mathrm{hop}(v)$:
> $$2\text{-}\mathrm{hop}(v):=({{\bullet}}_{ _{u\in N(v)}}S_u)\bullet S_v$$
>
> Here, $S_u$ and $S_v$ represent the monoidal elements corresponding to $N(u)$ and $N(v)$, the neighborhood of $u$ and $v$ (see the proof of theorem 2.5.1). The transformation $\mathsf{Tr}$ maps this cover into a collection of matrices, providing an alternative for adjacency matrix. This approach can be generalized to $k\text{-}\mathrm{hop}(v)$.
>
> $\textbf{Second Example:}$
>
> For a node $v \in G$, we select the highest-degree neighbors $v^1, \ldots, v^t \in N(v)$ and form the monoidal element $D_1^v$:
> $$D_1^v = e_1 \bullet \cdots \bullet e_t$$
>
> where $e_i$ represents the directed edge from $v$ to $v^i$. In subsequent iteration, we exclude $v$ and recursively construct elements $D_1^{v^1}, \ldots, D_1^{v^t}$. After $k$ iterations, we aggregate the results to form $D_v$:
> $$D_v = D_1^v \bullet \cdots \bullet D_k^v$$
>
> The transformation $\mathsf{Tr}$ converts these elements into matrix representations, offering an alternative for adjacency matrix that incorporates information about higher-degree nodes.
>
> These examples highlight how the GGNN framework translates diverse ideas into implementable GNN models, enabling flexible exploration of graph structures.
>
> $\textbf{2. Expressivity and Comparisons with}$ $k$-WL
>
> As demonstrated experimentally, $\mathsf{SNN}$ surpasses $3$-WL in distinguishing strongly regular graphs. It is empirically verified that $\mathsf{SNN}$ can resolve structural differences that $3$-WL cannot. We acknowledge the importance of theoretical analysis and are actively working on formalizing comparisons between $\mathsf{SNN}$ and $k$-WL in an ongoing study.
>
> $\textbf{3. Clarification of Main Contribution}$
>
> While $\mathsf{SNN}$ exemplifies the GGNN framework in practice, our main contribution lies in the algebraic tools provided by GGNN. These tools facilitate the definition of covers, the use of monoidal structures, and the transformation $\mathsf{Tr}$, enabling systematic development of novel GNN architectures. This framework helps to get rid of conventional neighborhoods.
>
> $\textbf{4. Scalability and Performance}$
>
> Most real-world graphs are sparse. As discussed in the paper, $\mathsf{SNN}$ has a time complexity of $\mathcal{O}(|E| \cdot |V|^2)$. In practical scenarios, as outlined in our experiments, $\mathsf{SNN}$ is primarily used to preprocess the dataset, transforming it into a new representation. Subsequently, any message-passing operator can be applied during the training phase, ensuring scalability and compatibility with existing frameworks.
>
> We sincerely hope these clarifications address your concerns and strengthen the understanding of our contributions. Thank you again for your valuable comments.

---

> > ### Author Response · Authors · 2024-11-22
> >
> > We thank the reviewer for the insightful suggestion to evaluate our model on the BREC benchmark, a more fine-grained expressivity benchmark.
> >
> > $\textbf{Results Summary on the BREC Benchmark}$
> >
> > We evaluated the performance of $\mathsf{SNN}$ on four categories of graphs in the BREC benchmark. The number of distinguished graph pairs for each category is as follows:
> >
> > In $\textbf{Basic Graphs}$ (60 pairs), $\mathsf{SNN}$ distinguished all 60 pairs.
> >
> > In $\textbf{Regular Graphs}$ (140 pairs), $\mathsf{SNN}$ distinguished all 140 pairs.
> >
> > In $\textbf{Extension Graphs}$ (100 pairs), $\mathsf{SNN}$ distinguished all 100 pairs.
> >
> > In $\textbf{CFI Graphs}$ (100 pairs), $\mathsf{SNN}$ distinguished 65 pairs.
> >
> > Based on our knowledge, the results achieved by our method are currently the best reported in the literature on this benchmark. This demonstrates the effectiveness of our approach in distinguishing between different graphs and further strengthens the contribution of our work.

---

> > > ### Comment · Reviewer_kNJA · 2024-11-25
> > >
> > > I sincerely thank the authors for their efforts. The responses basically address my concerns, and the results on BREC is convincing. I encourage the authors to include these new results in their new version of the paper. I will keep my positive rating.
> > >
> > > A minor point: can you please use markdown instead of latex in your response? The openreview platform only supports the former.

---

### Author Response · Authors · 2024-11-20

Respectfully, we believe that providing further clarification regarding the objectives of this paper will enhance its clarity. Therefore, in the following, we offer a more detailed explanation of the contributions of the paper.

$\textbf{Main Contribution Clarification}$

$\textbf{1. Introducing the Concept of Covers as an Extension of Neighborhoods}$

In $\textbf{Definition 2.2.1}$, we introduce the monoid $(\mathsf{Mod}(G), \bullet)$ as the foundation for the concept of neighborhoods.

In $\textbf{Definition 2.2.2}$, we define Covers as an alternative to neighborhoods, offering a broader perspective.

In $\textbf{Theorem 2.5.1}$, we prove that neighborhoods can serve as covers for graphs, thereby establishing a direct connection between these concepts.

$\textbf{2. Transforming Covers into Collections of Matrices as Alternatives to the Adjacency Matrix}$

In $\textbf{Definition 2.3.1}$, we present the monoid $(\mathsf{Mom}(G), \circ)$, which serves as the domain for matrix representations of the elements in $(\mathsf{Mod}(G), \bullet)$.

In $\textbf{Theorem 2.3.3}$, we introduce the monoidal homomorphism $\mathsf{Tr}$, which acts as a bridge between $(\mathsf{Mod}(G), \bullet)$ and $(\mathsf{Mom}(G), \circ)$. Notably, $\mathsf{Tr}$ respects binary operations, meaning $\mathsf{Tr}(D_1 \bullet D_2) = \mathsf{Tr}(D_1) \circ \mathsf{Tr}(D_2)$. This property is crucial for computing matrix interpretations of covers.

$\textbf{3. The Grothendieck Graph Neural Networks (GGNN) Framework}$

In $\textbf{Definition 2.5.1}$, we introduce the GGNN framework, which is not a specific GNN model but a platform for developing GNN models. The tools provided within this framework (see $\textbf{Theorem 2.2.2}$ and $\textbf{Proposition 2.4.1}$) make it a comprehensive ``studio'' for crafting GNN models.

$\textbf{Theorems 2.4.1}$ and $\textbf{2.4.2}$ demonstrate that the GGNN framework offers an algebraic description of graphs that is unique up to isomorphism. This significantly enhances the design of invariant GNN models.

$\textbf{Secondary Contribution}$

$\textbf{1. $\mathsf{SNN}$ as a Topology-Aware GNN Model}$

We introduce $\mathsf{SNN}$ as an example of a model built within the GGNN framework. This model is specifically designed to capture and utilize the topological structure of graphs.

As shown in the experiments, $\mathsf{SNN}$ effectively distinguishes all graphs within the available collections of Strongly Regular Graphs, demonstrating its capability in topology-aware graph classification.

$\mathsf{SNN}$ provides a complete example of how the GGNN framework can be employed to construct GNN models.

---

### Author Response · Authors · 2024-11-25

Dear Reviewers,

we respectfully ask if you could review our responses to your comments and update your evaluation of our work.

Thank you once again for your time and feedback.

Best regards,

---

### Author Response · Authors · 2024-11-29

Dear Reviewers, we have decided to withdraw our paper. We sincerely appreciate the time and effort you have invested in reviewing our work.

---

### Note · Authors · 2024-11-29

I have read and agree with the venue's withdrawal policy on behalf of myself and my co-authors.